# The Molecular Aspects of Functional Activity of Macrophage-Activating Factor GcMAF

**DOI:** 10.3390/ijms242417396

**Published:** 2023-12-12

**Authors:** Svetlana S. Kirikovich, Evgeniy V. Levites, Anastasia S. Proskurina, Genrikh S. Ritter, Sergey E. Peltek, Asya R. Vasilieva, Vera S. Ruzanova, Evgeniya V. Dolgova, Sofya G. Oshihmina, Alexandr V. Sysoev, Danil I. Koleno, Elena D. Danilenko, Oleg S. Taranov, Alexandr A. Ostanin, Elena R. Chernykh, Nikolay A. Kolchanov, Sergey S. Bogachev

**Affiliations:** 1Institute of Cytology and Genetics of the Siberian Branch of the Russian Academy of Sciences, 630090 Novosibirsk, Russia; levites@bionet.nsc.ru (E.V.L.); likhacheva@bionet.nsc.ru (A.S.P.); ritter@bionet.nsc.ru (G.S.R.); peltek@bionet.nsc.ru (S.E.P.); vasilieva@bionet.nsc.ru (A.R.V.); ruzanova@bionet.nsc.ru (V.S.R.); dolgova.ev@mail.ru (E.V.D.); s.oshikhmina@g.nsu.ru (S.G.O.); kol@bionet.nsc.ru (N.A.K.); 2N.N. Vorozhtsov Novosibirsk Institute of Organic Chemistry of the Siberian Branch of the Russian Academy of Sciences, 630090 Novosibirsk, Russia; avs_tur1987@mail.ru (A.V.S.); koleno_danil@mail.ru (D.I.K.); 3State Research Center of Virology and Biotechnology “Vector”, 630559 Koltsovo, Russia; danilenko_ed@vector.nsc.ru (E.D.D.); taranov@vector.nsc.ru (O.S.T.); 4Research Institute of Fundamental and Clinical Immunology, 630099 Novosibirsk, Russia; ostnin62@mail.ru (A.A.O.); ct_lab@mail.ru (E.R.C.)

**Keywords:** vitamin D3 binding protein, group-specific component protein-derived macrophageactivating factor, CLEC10A, pro-inflammatory and anti-inflammatory cytokines

## Abstract

Group-specific component macrophage-activating factor (GcMAF) is the vitamin D_3_-binding protein (DBP) deglycosylated at Thr^420^. The protein is believed to exhibit a wide range of therapeutic properties associated with the activation of macrophagal immunity. An original method for GcMAF production, DBP conversion to GcMAF, and the analysis of the activating potency of GcMAF was developed in this study. Data unveiling the molecular causes of macrophage activation were obtained. GcMAF was found to interact with three CLEC10A derivatives having molecular weights of 29 kDa, 63 kDa, and 65 kDa. GcMAF interacts with high-molecular-weight derivatives via Ca^2+^-dependent receptor engagement. Binding to the 65 kDa or 63 kDa derivative determines the pro- and anti-inflammatory direction of cytokine mRNA expression: 65 kDa—pro-inflammatory (TNF-α, IL-1β) and 63 kDa—anti-inflammatory (TGF-β, IL-10). No Ca^2+^ ions are required for the interaction with the canonical 29 kDa CLEC10A. Both forms, DBP protein and GcMAF, bind to the 29 kDa CLEC10A. This interaction is characterized by the stochastic mRNA synthesis of the analyzed cytokines. Ex vivo experiments have demonstrated that when there is an excess of GcMAF ligand, CLEC10A forms aggregate, and the mRNA synthesis of analyzed cytokines is inhibited. A schematic diagram of the presumable mechanism of interaction between the CLEC10A derivatives and GcMAF is provided. The principles and elements of standardizing the GcMAF preparation are elaborated.

## 1. Introduction

The macrophage-activating factor GcMAF belongs to the category of plasma proteins, which have a wide range of properties and significant therapeutic potential [1,2,3,4,5,6,7,8,9]. Different research teams have reported the anticancer activity of the macrophage-activating factor [10,11,12,13,14,15,16,17,18,19]. GcMAF was shown to have an ability to rectify neurodegenerative diseases including autism spectrum disorder [7,20,21,22,23]. The conducted experiments demonstrate that the wide range of therapeutic properties of GcMAF is directly related to its impact on the macrophage [8,9,24,25,26]. It is believed that the GcMAF precursor, vitamin D_3_-binding protein, is converted to the macrophage-activating factor via the partial deglycosylation of trisaccharide linked to Thr^420^ by the O-glycosidic bond, giving rise to the terminal GalNAc residue. The available experimental data indicate that this very carbohydrate is responsible for the macrophage-activating ability of GcMAF [11,27,28,29,30,31]. Some preparations are called GcMAF, but, in fact, are not GcMAF. Those are examples of the so-called second-generation GcMAF, which actually is a mixture of enzyme-treated plasma proteins, and the so-called third-generation GcMAF, which is colostrum containing terminal, unbound GalNAc [26,32,33,34]. In our early studies of GcMAF, we faced the complicacy of the procedure of isolating DBP from plasma and converting it to GcMAF [11,21,35,36]. In the present study, we elaborated the original method for both producing and converting GcMAF as well as analyzing its activation potency. Simultaneously, we formulated the principles and elements for standardizing the GcMAF preparation. The production of the GcMAF preparation has been patented (priority No. 47390 2023121663 of 17 08 2023).


**Technical note. Study purpose and general design.**


In the present study, we attempted to identify the molecular elements that determine the pro- or anti-inflammatory responses of GcMAF-activated peritoneal macrophages (PMs).

The paper includes two main parts of results, a discussion, and a general conclusion.

The first part of the study results section includes the following stages:(1)Proteins binding to the 25-OH/Sepharose^®^ resin through an affine site have been characterized;(2)Standardization elements for GcMAF isolation and analysis, as well as the assessment of the specific activity of PMs treated with GcMAF preparations obtained using various conversion methods, have been developed;(3)The activation of the mRNA synthesis of inflammatory mediators IL-1b and TNF-α and major anti-inflammatory cytokines TGF-β and IL-10 in PMs isolated from C57BL|6 mice via DBP preparations from various donors was assessed.(4)The activation of the mRNA synthesis of IL-1b, TNF-α, TGF-β, and IL-10 in PMs through GcMAF preparations obtained through the conversion of leukocyte precursors activated using various treatment modes was assessed.(5)Fluorescence microscopy analysis of the distribution of Cy5-labeled GcMAF on PMs was conducted.

The conclusion to the first part summarizes the main findings of the experimental work.

The second part of the study results section includes the following stages:(1)A possible mechanism of macrophage activation by GcMAF was analyzed based on the literature data.(2)Sandwich-type blot assay was developed, which made it possible to characterize the interaction of GcMAF and the purified commercial protein CLEC10A.(3)Using the approach mentioned in the previous paragraph, the interaction between GcMAF and CLEC10A in mouse PMs was analyzed.

The conclusion to the second part summarizes basic molecular elements and formulates the principle of macrophage activation towards either pro- or anti-inflammatory response.

The Discussion section analyzes the onset mechanisms of PM inflammatory response and its inhibition in response to the treatment of phagocytes with GcMAF.

The Conclusion provides a hypothetical scheme of interaction between GcMAF and CLEC10A leading to the onset of pro- or anti-inflammatory response in mouse PMs.

## 2. Results

Part I.

### 2.1. Comparative Characterization of GcMAF-RF Produced by Affinity Chromatography on an Actin/Chitin and 25-OH/Sepharose^®^ Columns. Tandem Mass Spectrometry Analysis of Two Protein Fractions Isolated by Chromatography on a 25-OH/Sepharose^®^ Column

In our earlier studies [36], macrophage-activating factor GcMAF-RF was produced using vitamin D_3_-binding protein (DBP) after the affinity chromatography of donor-derived plasma on an Actin/chitin column followed by the conversion of DBP to the specific macrophage-activating factor GcMAF-RF by treating DBP with leukocyte suspension for 12–24 h in an atmosphere of CO_2_ (Figure 1A). The specificity of the resulting macrophage-activating factor was determined either via Western blotting (for anti-Gc antibodies) or in special experiments assessing the interaction with H. pomatia lectin [37]. Impurity proteins that also specifically interact with actin were always isolated along with the target protein in all the isolation experiments. Some of these major impurity fractions were characterized in other studies [11,29,38,39,40,41]. The target protein isolated using this procedure could not be standardized, which was required for performing detailed molecular studies and developing a medicinal product based on a specific macrophage-activating factor.

Therefore, a system for precursor isolation via affinity chromatography has been designed, employing 25-OH-D_3_/Sepharose^®^ as an affinity matrix. The matrix was produced in a way completely different from the method reported by [35], and its fabrication is subject to industrial ownership by the Open Joint-Stock Company “ACTIVATOR MAF”. The product obtained using this affinity chromatography system consisted of two proteins (Figure 1A left block). One of them migrated at the level identical to that for GcMAF previously reported in the literature and specifically interacted with anti-Gc antibodies. The other protein migrated at a level of ~28–30 kDa and did not interact with anti-Gc antibodies (Figure 1A right block).

An assumption was made, quite reasonable in our opinion, that sequential dual-affinity chromatography would yield a protein of the highest purity grade. Both variants of sequential chromatography (Actin/chitin—25-OH-D_3_/Sepharose^®^ and 25-OH-D_3_/Sepharose^®^—Actin/chitin) were analyzed. The results are shown in Figure 1B.

DNA and the amino acid sequence, as well as the functional domains of both DBP and GcMAF, were characterized in refs. [6,42,43,44]. The structure of the full-length and potentially the maximally functional protein contains the vitamin D_3_-binding and actin-binding domains as well as the domain responsible for interaction with neutrophils and the glycosylation site consisting of three saccharides and comprising N-acetyl galactosamine (GalNAc) attached to Thr^420^ via the O-glycosidic bond, in addition to terminal sialic acid and galactose covalently bound to GalNAc (Figure 1D). Along with the numerous allelic forms [44,45,46], forms (variants) of the native protein with a reduced number of functional groups and smaller degree of glycosylation can exist in the human body. This idea has been supported by findings obtained in the studies focusing on site-specific glycosylation where different variants of glycosylation, positions of saccharide residues, and the potentiality of several saccharide residue attachment sites have been detected [4,27,47,48,49] (Figure 1(D4)). In addition, protein conformation is important for specific interaction; its change can lead to either complete or partial affinity deterioration [30].

The conducted analysis indicated that two different forms of DBP are isolated via affinity chromatography in the case of two different domains. If the 25-OH-D_3_/Sepharose^®^ column is the primary one, then ≤10% of MAF obtained from the initial amount of DBP eluted from the 25-OH-D_3_/Sepharose^®^ column binds to the Actin column. If the primary column is the Actin column, a trace amount (<1%) of initial protein binds to the 25-OH-D_3_/Sepharose^®^ matrix. This means that (1) most of the DBP removed from the 25-OH-D_3_/Sepharose^®^ column contains no actin-binding domain, or the corresponding amino acid sequence or protein conformation is functionally altered, and (2) most of the DBP removed from the actin/chitin column does not contain the D_3_-binding domain (Figure 1B). The described part of the study was carried out using plasma derived from donor LEV (C GcMAF LEV L pure 37 °C). All the experiments described in our studies [36,37,50,51] were conducted using biological material obtained from this very donor. The precursor was converted to DBP using the leukocyte suspension derived either from the same donor or from different donors (the activating ability of leukocytes is characterized below and in Figure 2A).

A discretionary decision to use the product obtained on the 25-OH-D_3_/Sepharose^®^ column for further studies has been made, with the following reasoning: (1) it is a stable chemically modified matrix that allows many chromatography runs to be performed; and (2) the resulting chromatography product is homogeneous and always consists of the two proteins described below and, therefore, it is possible to perform further molecular studies using the macrophage-activating factor with the standardized isolation procedure.

We performed the chromatographic separation of the components of bands using HPLC on a Thermo Fisher Scientific Ultimate 3000 Series system followed by the identification of the compounds on an Orbitrap Fusion Lumos tandem mass spectrometer. Electronic sputtering ionization made it possible to produce multi-charged ions whose advantage was that secondary-ion mass spectra could be recorded upon the fragmentation of precursor ions with different charges, which increases the level of amino acid sequence identification. The Proteome Discoverer 2.4 software revealed a 63.66% identity between the upper band (58 kDa) and the vitamin D_3_-binding protein, which was sufficient for identifying the product obtained on the 25-OH-D_3_/Sepharose^®^ column as DBP. This result allowed us to use the term conventionally employed to denote the macrophage-activating factor as GcMAF, without adding the words “related factor” (RF), in our further studies. The short protein (28 kDa) shows 74.54% identity with apolipoprotein A1 (ApoA1) (Figure 1C), which is sufficient for identifying this protein as ApoA1.

We carried out sandwich-type dot blot assay to assess the feasibility of direct interaction between the precursor and ApoA1, where the product of affinity chromatography on the 25-OH-D_3_/Sepharose^®^ column fractionated in gel and transferred to the membrane was initially treated with DBP or GcMAF, and the complex was subsequently visualized using anti-Gc antibodies. Under the selected conditions, DBP did not directly interact with ApoA1. However, GcMAF bound to ApoA1 (Figure 1(E1), shown with an arrow).

Therefore, it turned out that protein variants with different molecular characteristics can be isolated on an affinity chromatography matrix. Additionally, it was found that ApoA1, as a single factor, does not affect the synthesis of the analyzed PM cytokines (Appendix A).

### 2.2. Elaborating the Standardization Elements of DBP and GcMAF Production and Assessing the Specific Activity of PMs Treated with GcMAF Preparations Obtained Using Different DBP Conversion Methods

This finding and further studies allowed us to outline the elements of standardizing DBP and GcMAF:(1)Isolating the factor on an original 25-OH-D_3_/Sepharose^®^ column;(2)Using the PM cell system obtained from C57BL/6 mice, which was selected for analysis. It is a genetically homogeneous mouse line, which allows employing such animals as a standard source of PMs; and(3)Employing the ability of DBP and GcMAF to induce the mRNA synthesis of four cytokines (pro-inflammatory mediators TNF-α and IL-1β as well as the major anti-inflammatory cytokines TGF-β and IL-10) in PMs, which was selected for analysis.

In this study, we assessed four variants of treating leukocytes used for converting DBP to GcMAF.

The first variant is referred to as the conventional mode. DBP is isolated on the column, eluted, and converted to GcMAF using treatment with leukocyte suspension in PBS supplemented with 10% FBS. Next, depending on the objective set, either the macrophage-activating factor is used for the experiments without any treatment, or serum proteins and metabolic by-products of leukocytes are removed from it through repetitive chromatography [27]. In this study, a rapid and efficient method for converting DBP to GcMAF directly on the resin was developed, which was dubbed rapid isolation. In this case, all the treatments of DBP are conducted when the protein is associated with the matrix.

In the second variant, DBP immobilized on the resin is treated with leukocyte suspension in native plasma at 37 °C.

In the third variant, DBP immobilized on the resin is treated with leukocyte suspension in native plasma in the presence of LysoPC [47].

In the fourth variant, DBP immobilized on the resin is treated with leukocytes washed in buffer A (see the Materials and Methods section), which are activated towards inflammation with LysoPC for 3 h at 37 °C in the presence of 2 mM Ca^2+^.

GcMAF specimens obtained after these treatments were washed to remove the conversion medium, eluted with 3M guanidine chloride, dialyzed against PBS, and used for activating PMs.

The comparative analysis of the protein (C GcMAF LEV L pure 37 °C) produced by conversion in leukocyte suspension and using the rapid isolation procedure revealed that both specimens had equal activating abilities. It was assumed and demonstrated later that under identical conditions, DBP can be converted to GcMAF by treating with sialidase and β-galactosidase in the solution, which considerably simplifies the procedure of isolating the macrophage-activating factor.

(4)Therefore, the fourth element in standardizing the procedure of GcMAF production is converting DBP to macrophage-activating factor directly on 25-OH-D_3_/Sepharose^®^.

Treatment with leukocyte suspension implied that plasma would necessarily contain the metabolic by-products of leukocytes. In some variants, the conversion medium (PBS) also contained 10% FBS. It was demonstrated earlier that the conversion medium (PBS, 10% FBS) per se had no effect on the phagocytic activity of PMs [36]. This meant that it was DBP converted to GcMAF that activated PMs. We needed to additionally assess the effect of the conversion medium on the parameters analyzed in this study, namely, mRNA synthesis of both pro-inflammatory cytokines TNF-α and IL-1β as well as the major anti-inflammatory cytokines TGF-β and IL-10 in PMs. The treatment of PMs with the conversion medium free of DBP but containing leukocytes not induced toward inflammation has no effect on the synthesis of the aforementioned cytokines in the PMs (Figure 2A). This suggests that leukocytes per se are not capable of activating PMs.

Therefore, the model for assessing the synthesis of pro-inflammatory mediators TNF-α and IL-1β and the major anti-inflammatory cytokines TGF-β and IL-10 by peritoneal macrophages isolated from C57BL/6 mice was chosen to analyze the activating ability of DBP and GcMAF. The GcMAF preparation produced using affinity chromatography on the 25-OH-D_3_/Sepharose^®^ column, which was converted either in the solution or directly on the resin by different conversion media containing donor-derived leukocytes, was used.

Different modes of precursor conversion using leukocytes were analyzed in this study. Furthermore, the results of DBP conversion through treatment with sialidase and β-galactosidase enzymes were used for comparison. We have systematized these modes and used the respective denotations. Isolation under the conventional conditions—c; isolated on resin—Ex (express (rapid) method); leukocytes—L; in plasma—plasma; using LysoPC—LysoPC; purified leukocytes—pure; leukocytes were treated at different temperatures: 37 °C and 39 °C.

### 2.3. Quantifying Activation of mRNA Synthesis of Both Pro-inflammatory Mediators TNF-α and IL-1β as well as the Major Anti-Inflammatory Cytokines TGF-β and IL-10 in PMs Isolated from C57BL/6 Mice Using DBP Preparations Derived from Different Donors

At the initial analysis stage, we found that DBP derived from several (four) donors activates PMs toward synthesizing the mRNA of one or several cytokines that, in some cases, refer to differently oriented inflammatory responses (Figure 2B–E).

DBP LEV insignificantly activated PMs toward IL-1β synthesis (we would like to mention, once again, that this very precursor was used in all the experiments in all our previous studies [36,37,50,51]). It activated PMs towards synthesizing anti-inflammatory cytokines (Figure 2B).

DBP1 activated the mRNA synthesis of TNF-α, IL-1β, and TGF-β, which refer to the oppositely oriented inflammatory responses.

DBP2 activated the mRNA synthesis of IL-1β and had an insignificant effect on the mRNA synthesis of anti-inflammatory cytokines.

DBP3 activated the mRNA synthesis of TNF-α and IL-1β in a dose-dependent manner. For all the three doses of the activator, the level of mRNA synthesis of TGF-β was approximately six times higher than the control values.

### 2.4. Quantifying Activation of mRNA Synthesis of Both Pro-inflammatory Mediators TNF-α and IL-1β As Well as the Major Anti-Inflammatory Cytokines TGF-β and IL-10 in PMs Isolated from C57BL/6 Mice Using GcMAF Preparations Obtained via Precursor Conversion Using Leukocytes Activated in Different Treatment Modes

A comprehensive comparative analysis of the ability of precursor preparations and GcMAF based on them to activate the mRNA synthesis of selected cytokines was conducted. We compared different doses, temperatures, conversion methods, and leukocyte suspensions prepared using different methods. Three doses (0.02 µg, 0.2 µg, and 2 µg) were used in all the experiments; in some experimental runs, we employed the dose of 10–20 µg target protein per point (10^6^ PMs in 0.5 mL of DMEM complete cell culture medium). The results are presented in the logic diagram below. Figure 3 and Figure 4 show the results of all the experiments. The results of analyzing both the mRNA synthesis of the studied cytokines as well as the phagocytic activity of the same macrophages are shown for preparations C GcMAF LEV L pure 37 °C and GcMAF2. The findings were classified according to DBP used in the study.

We considered C GcMAF LEV L pure 37 °C. An analysis was carried out using three model systems—the J774 macrophage cell line, PMs treated ex vivo, and experimental animals treated in vivo—followed by the analysis of PMs. For the cell culture, it was found that IL-1β mRNA synthesis was activated. mRNA synthesis of other cytokines was not activated. The ex vivo treatment of PMs activated the synthesis of all the analyzed cytokines. For the in vivo system, the mRNA synthesis of pro-inflammatory cytokines was insignificant. For the in vivo treatment, the synthesis of anti-inflammatory TGF-β and IL-10 increased tenfold compared to the control. The in vivo analysis was indicative that PMs subjected to activation with C GcMAF LEV L pure 37 °C exhibited an anti-inflammatory response.

An analysis of the diagram of phagocytic activity after activation with C GcMAF LEV L pure 37 °C at different doses showed that activity increased sequentially with an increasing preparation dose. The mRNA synthesis of anti-inflammatory cytokines TGF-β and IL-10 reached the maximal values at two minimal doses and dropped to zero at the next maximally selected dose of 1–2 µg (Figure 3A–C,E–G). Therefore, anti-inflammatory cytokines TGF-β and IL-10 are characterized by a non-monotonic (bell-shaped) diagram for mRNA synthesis with the maximum being at low doses (0.02, 0.2 µg). For all the analyzed cytokines, the value corresponding to the dose of 1–2 µg dropped to zero. Figure 3E–G present a comparative analysis of PMs’ ability to induce anti-inflammatory cytokine mRNA synthesis with the simultaneous phagocytosis of metallic beads. These PMs are likely to have the M2 phenotype. Graph (G) shows that anti-inflammatory cytokine mRNA synthesis does not interfere with the PM phagocytic activity, which remains at a high level. After the inhibition of anti-inflammatory cytokine mRNA synthesis by PMs, which is believed to be characteristic of the M2 phenotype, the phagocytic activity increases significantly.

The maximum pro-inflammatory response of phagocytes was attained when PMs were activated with Ex GcMAF LEV L pure LysoPC 37 °C converted by pure leukocytes obtained by performing rapid treatment with LysoPC (Figure 3D).

GcMAF2 elicits a pro-inflammatory response in PMs (Figure 4C). Figure 4D–F present a comparative analysis of the ability of phagocytes to induce the pro-inflammatory response with the simultaneous internalization of metallic beads. These macrophages are likely to have the M1 phenotype. The graph demonstrates that the mRNA synthesis of pro-inflammatory cytokines TNFα и IL-1b does not interfere with the PM phagocytic activity, which remains at a high level. In the case of an excess of ligand in relation to PMs, which is believed to be characteristic of the M1 phenotype, mRNA synthesis of these cytokines decreases. In addition to the decrease in TNFα and IL-1b mRNA expression, the phagocytic activity of PMs also decreases.

The obtained result of comparative PM activation by two variants of the GcMAF activator indicates the following. In case the M2 phenotype of PMs is activated, the phagocytic activity is present as a normal phagocyte response. When TGF-β and IL-10 mRNA synthesis is inhibited, phagocytic activity increases. If the M1 phenotype of PMs is activated, the phagocytic activity is at a high level and reduces together with a decrease in the mRNA expression level of pro-inflammatory cytokines.

Thus, phagocytic activity as a normal reaction of phagocytes is found in both M1 and M2 PMs. Differences are observed when mRNA synthesis is inhibited. In the case of M2 cells and the inhibition of anti-inflammatory cytokine expression, the phagocytosis is significantly enhanced. In the case of M1 cells, in addition to a decreased expression of pro-inflammatory cytokines, the phagocytic activity of the PMs is also reduced.

GcMAF1 elicits a pro-inflammatory response in PMs. The response is maximal when using LysoPC (Figure 4A,B).

GcMAF3 elicits an oscillatory response in PMs depending on treatment with leukocytes. One of the selected modes inhibits the mRNA expression of all the analyzed cytokines (Figure 4G). The other two modes exhibit multi-directional effects in PM activation (Figure 4H,I).

### 2.5. Fluorescence Microscopy Analysis of Cy5-Labeled GcMAF Distribution on PMs

In the experiments using C GcMAF LEV L pure 37 °C, we observed a non-monotonic (bell-shaped) curve for the efficiency of mRNA synthesis and a well-defined effect of inhibiting the synthesis of any cytokines when using GcMAF at a dose of 2 µg. This phenomenon has been reported in the literature and is believed to be related to receptor aggregation at high ligand doses. It implied that cytology analysis of a specimen with a large amount of the ligand may show the areas of receptor molecule aggregation on PMs [52,53,54,55]. For testing this assumption, C GcMAF LEV L pure 37 °C was labeled with Cy5, and the treatment of PMs was performed. Three working doses were selected (0.02 µg, 0.2 µg, and 2.0 µg). Live-cell confocal imaging demonstrated that for the two smaller doses, the labeled material was located in the cytoplasm and the cell nucleus (Figure 5A–C). In the case of using the 2.0 µg dose, (1) a very strong-intensity label covering the entire cell, (2) an intracellular label, and (3) specific fluorescence spots on the macrophage surface were detected (Figure 5D). These spots can be aggregated receptors structured by excess ligand, which agrees with the opinion available in the literature.

Conclusions of the first study part.

The analysis conducted in the first part of the study revealed the following GcMAF/PM system features.

(1)Tandem mass spectrometry analysis showed that DBPs are eluted from the affinity column together with the protein ApoA1. The sandwich-type assay suggests that GcMAF (but not DBP) can directly interact with ApoA1. However, we believe that the joint elution of the two factors is associated with a ligand molecule, one part of which acts as an affinity matrix for GcMAF and the other part of which serves as a matrix for ApoA1;(2)DBPs from various donors without additional treatment activate PMs to produce the analyzed cytokines;(3)Phagocytic activity as a normal phagocyte response is observed in both M1 and M2 PMs. Differences are observed upon the inhibition of the mRNA synthesis. In the case of M2 cells and the inhibition of anti-inflammatory cytokine expression, phagocytosis is significantly enhanced. In the case of M1 cells, in addition to the decreased expression of pro-inflammatory cytokines, the phagocytic activity of PMs is also reduced;(4)High doses of GcMAF (>1 μg in the working system, see Materials and Methods) completely inhibit the expression of the analyzed cytokines. This phenomenon is believed to be associated with receptor aggregation on the membrane after ligand binding (GcMAF);(5)Using the DBP to GcMAF conversion approach with the involvement of leukocytes, as described by Yamamoto [56], we attempted to find a mode that could be reproduced in all subsequent experiments. We did not manage to clearly determine the causes of pro- and anti-inflammatory response in PMs using this approach.

This conclusion determined the need to continue the search.

Part II.

### 2.6. The Possible Mechanism of PM Activation by GcMAF

As it follows from the experimental research literature, GcMAF carries a free GalNAc moiety at the Thr^420^ position of the amino acid sequence [6,11,28,29,30,31,42,44,45,46]. The high-affinity carbohydrate-binding receptors have been studied well [57]. Two specific receptors have been described for GalNAc: ASGR1 (CLEC4H1) and CLEC10A (MGL or CD301). In the absence of pathological manifestations, CLEC10A is expressed on tolerogenic tissue-resident dendritic cells, skin and lung macrophages, and PMs. Various inducing events significantly increase the expression of C-lectin receptor; upon engagement with the ligand, tolerogenic antigen-presenting cells induce either the development of Tregs (regulatory leukocytes) or the anergy of immune cells and, in particular, T cells via an MGL (CLEC10A)-dependent mechanism [57,58,59,60]. This means that CLEC10A is the main candidate for interacting with GcMAF. The fact that DBP (whose GalNAc is supposed to be closed by residues of two other saccharides) induces the mRNA synthesis of various analyzed cytokines in certain cases, in either a dose-dependent or dose-independent manner, may indicate that there are three possible scenarios. (1) Some DBPs have already been deglycosylated and carry terminal GalNAc. However, the activation curves are supposed to coincide with the curves of GcMAF in this case, but they differ substantially (Figure 2, Figure 3 and Figure 4). (2) Partial deglycosylation is possible. (3) mRNA synthesis in PMs is activated by DBP via a different mechanism.

Having compared the mRNA synthesis curves influenced by DBP and GcMAF, one can assume that the treatment of DBP with leukocytes alters the DBP structure so that the activation mode changes completely. DBP binds to a certain macrophage-activating factor and activates PMs. In our research, conversion modified the DBP molecule and either cancelled the previously existing binding between DBP and PMs or formed a new one, which altered the activation mode.

Studies characterizing the interaction between DBP, GcMAF, and CLEC10A were conducted to shed light on these aspects.

### 2.7. Analysis of the Interaction between GcMAF and Commercially Purified CLEC10A

GcMAF contains a ring-opened or partly ring-opened adduct N-acetylgalactosamine (GalNAc) bound to Thr^420^. The deglycosylation of trisaccharide and removal of sialic acid and(or) galactose (as an example) at the inflammation site is a principal step in converting vitamin D3-binding protein (DBP) to the macrophage-activating factor (GcMAF) [27,56,61].

Properties of specific carbohydrate-binding receptors (lectins) have been intensively studied for a long time; with a member belonging to the large family of C-type Ca^2+^-dependent lectin receptors, namely CLEC10A (the macrophage galactose type lectin, MGL, CD301), being one of them. CLEC10A carries a lectin recognition domain that binds the terminal or free GalNAc [60,62,63,64,65]. It is abundantly present on the plasma membrane of dendritic cells and activates macrophages M2 in a trimeric form [52,66,67,68,69,70]; according to all the current views, it is supposed to bind to GalNAc on the threonine^420(418)^ residue in GcMAF converted at the site of inflammation. This means that CLEC10A is a key candidate for playing the role of a factor that mediates the activating activity of GcMAF [37].

The sandwich-type assay procedure was developed, and a series of experiments characterizing the interaction between GcMAF and CLEC10A were conducted. Figure 6 shows the results of analyzing the interaction between these factors.

It was demonstrated that PMs of C57BL/6 mice contain CLEC10A and that the receptor specifically interacts with the macrophage-activating factor. Several fractions interacting with anti-CLEC10A antibodies were detected in the blot (Figure 6A,B).

An analysis of the cross-reactivity of anti-GC and anti-CLEC10A antibodies revealed no significant cross-affinity (Figure 6C,D), so the experiments focusing on direct interaction between GcMAF and CLEC10A proteins could be carried out (Figure 6D lane 3).

In early experiments aiming to assess the direct interaction between CLEC10A and GcMAF exhibiting a stronger pro-inflammatory activity, we analyzed the potential interaction between the analyzed factors via formaldehyde crosslinking. Technically, the assay was carried out using the following procedure. Protein crosslinking in a solution was performed, followed by electrophoresis and transfer to the membrane. The membrane was treated with anti-CLEC10A antibodies. It was found that a high-molecular-weight fraction appeared, with mobility presumably corresponding to that of the GcMAF/CLEC10A complex (shown with an arrow) (Figure 6D). Experiments characterizing the direct interaction between the two factors by sandwich-type assay were then performed. After electrophoresis, either CLEC10A or GcMAF was transferred to the membrane and treated with a counteragent. The resulting complex was detected using the respective HRP-conjugated antibodies. Figure 7A–C,E,H present the experimental results. Specific protein–protein interactions in both directions were revealed. GcMAF preparations produced by conversion in different modes were used in the experiments. GcMAF3 produced in the Ex GcMAF3 L plasma 39 °C mode was chosen to be the major analyzed preparation for which near-zero mRNA expression of all the analyzed cytokines was observed (Figure 3G). Other GcMAF preparations were used in some other experiments, and these were directly specified when describing the experimental results.

One of the key questions was related to the effect of Ca^2+^ ions on the interaction of the terminal carboxyhydrate of DBP and GcMAF with CLEC10A. It is believed that Ca^2+^ ions are responsible for the direction of phagocyte response [64,65,71]. DBP3 precursor and GcMAF3 converted using the rapid method by leukocytes in native plasma activated by heating (39 °C) with neutral or nearly neutral characteristics (the Ex GcMAF3 L plasma 39 °C mode, Figure 4G) were used for analysis. The strips with separated CLEC10A were treated with the macrophage-activating factor in the presence of Ca^2+^ at different concentrations (2 mM, 20 mM, and 100 mM). After incubation, the complexes were detected using anti-Gc antibodies. The CLEC10A fraction corresponding to the generally known 29 kDa protein form [72,73] was found to react with both DBP and GcMAF, with no apparent difference under the Western blotting conditions. Specific interaction was detected only with the high-molecular-weight preparation fractions. Two proteins (the high-molecular-weight CLEC10A derivatives, ~63 and ~65 kDa) specifically binding to GcMAF appeared in the CLEC10A specimen treated with GcMAF3 (the Ex GcMAF3 L plasma 39 °C mode) in the presence of 20 mM Ca^2+^ ions. In this experiment, the strongest interaction was observed for the ~65 kDa protein. After the next rise in Ca^2+^ concentration, there was almost no binding. The findings suggested that under the Western blot assay conditions, 20 mM Ca^2+^ ions open the receptor site binding to the terminal carbohydrate in GalNAc of GcMAF3 (the Ex GcMAF3 L plasma 39 °C mode), which is typical of CLEC10A (shown with an arrow) (Figure 7B). We additionally analyzed the efficiency of CLEC10A–GcMAF binding (the Ex GcMAF1 L pure LysoPC 37 °C mode); significant pro-inflammatory activity at two temperatures (37 °C and 39 °C) in the presence of optimal Ca^2+^ concentration was observed. Temperature had no effect on interaction specificity (Figure 7C). The 65 kDa protein was the main reacting protein. The efficiency of interaction of the 63 kDa protein was severalfold lower.

In order to test the hypothesis that GcMAF binds through the terminal GalNAc, we conducted experiments involving the competitive inhibition of GalNAc–GcMAF binding by free GalNAc. The results of the analysis are presented in Figure 7E–G. Free GalNAc was shown to compete with the terminal GalNAc in GcMAF for binding to the specific high-molecular-weight CLEC10A derivative, thus preventing it from forming a clearly defined complex with a protein, which manifested itself as emergence of a blurred smear detectable by antibodies (Figure 7E; lane 4).

In order to elucidate the fact that the blurred smear belongs to the GalNAc/receptor complex rather than the GcMAF/receptor complex, we analyzed the ability of anti-Gc antibodies to interact with pure GalNAc. Anti-Gc antibodies were shown to specifically bind GalNAc, thus indicating that the carbohydrate-carrying site of the GcMAF protein is an antigen and induces anti-GalNAc antibodies, and this very interaction is presumably detected as a smear when GcMAF is competitively replaced from its complex with the 65 kDa high-molecular-weight CLEC10A fraction in the presence of 20 mM Ca^2+^ by 10 µg of free GalNAc (Figure 7E,F).

Ex vivo experiments characterizing the competitive interactions between free carbohydrate and terminal carbohydrate within GcMAF were carried out. PMs were treated with GalNAc, GcMAF, and GalNAc + GcMAF, and real-time PCR was conducted. Both compounds were shown to compete for binding to the receptor, which is responsible for the activation of the mRNA expression of the analyzed cytokine genes (Figure 7G). In this experiment, the selected GcMAF (the Ex GcMAF3 L plasma 39 °C mode) activated PMs to express both pro- and anti-inflammatory cytokines. Furthermore, the results obtained ex vivo demonstrate that free GalNAc does not activate PMs to express the mRNA of the analyzed cytokines.

A comparative analysis of the interaction of CLEC10A with the two GcMAF preparations produced using different conversion methods, C GcMAF LEV L pure 37° and Ex GcMAF3 plasma LysoPC 37 °C, was additionally conducted. The first GcMAF preparation activates the multi-directional responses in PMs (Figure 3B), while the second one activates PMs mostly toward the anti-inflammatory response, although insignificant mRNA expression of TNF-α and IL-1β genes is observed at certain doses (Figure 4I). It turned out that the two types of responses (i.e., the two types of GcMAF) are characterized by different patterns of binding to two high-molecular-weight CLEC10A derivatives. The C GcMAF LEV L pure 37 °C preparation, which activates both directions of the inflammatory response in PMs, is characterized by both fractions having a stronger intensity of the 65 kDa band, while the Ex GcMAF3 plasma LysoPC 37 °C preparation, which mostly activates the anti-inflammatory response in PMs, is predominantly characterized by the 63 kDa band (Figure 7H). This fact can indicate that the interaction of GcMAF mostly (or only) with the 63 kDa CLEC10A derivate is responsible for activating the anti-inflammatory response as the only direction, while binding only to the 65 kDa band is responsible for activating the pro-inflammatory response as it follows from PM activation by Ex GcMAF1 L pure LysoPC 37 °C (Figure 4A and Figure 7D). The simultaneous interaction with two derivatives activates both directions of the inflammatory response in PMs. This result also provides grounds for assuming that the two high-molecular-weight CLEC10A derivatives (63 kDa and 65 kDa) are the close “allelic” forms of the same protein factor that specifically bind GcMAF.

### 2.8. Analysis of Interaction of GcMAF and CLEC10A in Mouse PMs

In the final part of the experimental study, we analyzed several variants of interaction between GcMAF and proteins of PM lysate in the sandwich-type Western blot system in the presence and absence of Ca^2+^ ions (both in the culture medium and in assay system solutions). Ex GcMAF3 L plasma LysoPC 37 °C (Figure 4I) was shown to interact with the three major fractions of PM lysate, whose mobilities corresponded to those for 29 kDa, ~63 kDa, and ~65 kDa CLEC10A. In the presence of Ca^2+^ ions, specific interaction with all the CLEC10A fractions became stronger (Figure 8A,B).

It can be noted that in the case of macrophage lysates, when Ca^2+^ ions appear in solution, GcMAF binds more actively to the 63 kDa protein compared to the pure CLEC10A preparation, which correlates with a more pronounced anti-inflammatory response of PMF shown for this GcMAF preparation (Figure 8C–E). The difference can be attributed to differences in the structure of the authentic and commercially available CLEC10A. It is also fair to assume that two GcMAF variants differing in the location of sugar residues on the threonine^420(418)^ or the degree of deglycosylation interact with the two protein fractions, and there exists a stochastic competition in terms of internalization between the allelic CLEC10A forms binding to a particular GcMAF form in each particular case. This interaction mode suggests that there is a certain probability in eliciting the pro- or anti-inflammatory response in PMs, with the anti-inflammatory response being predominant for the analyzed GcMAF preparation. It should be noted that in this experiment, the appearance of calcium ions in the incubation medium significantly enhanced the connections between GcMAF and the canonical CLEC10A 29 kDa.

In addition to the results reported above, it was found that the total GcMAF, converted from a mixture of DBPs derived from different donors using purified sialidase and β-galactosidase enzymes via the rapid method, interacted mostly with the ~63 kDa CLEC10A fraction and activated PMs to synthesize only the anti-inflammatory IL-10. Meanwhile, in the same experiment, another preparation, Ex GcMAF LEV L pure LysoPC 37 °C, exhibiting a strong pro-inflammatory and simultaneously an anti-inflammatory activating effect on PMs, interacted with both ~63 kDa and ~65 kDa CLEC10A fractions in the sandwich-type blot assay (shown with arrows) (Figure 8C–E).

Conclusions of the second study part.

It is fair to say that PMs carry several derivatives of the CLEC10A receptor on their surface. Binding between GcMAF and the low-molecular-weight CLEC10A derivative (29 kDa) is necessary for ligand fixation. The interaction between GcMAF and two high-molecular-weight CLEC10A derivatives (63 kDa and 65 kDa) determines the direction of the inflammatory response. The pro-inflammatory response only is elicited in the case of ligand binding to the 65 kDa derivative; the anti-inflammatory response only is elicited in the case of ligand binding to the 63 kDa derivative; both directions of the inflammatory response in PMs are activated in the case of binding to both derivatives.

## 3. Discussion

### 3.1. Differences in DBPs Isolated Using the Actin/chitin and 25-OH-D_3_/Sepharose^®^ Affinity Sorbents, Tandem Mass Spectrometry Analysis of the Two Fractions Isolated Using 25-OH-D_3_/Sepharose^®^

This study attempted to characterize the activating impact of the macrophage-activating factor GcMAF on PMs in C57BL/6 mice. The results indicate that the macrophage-activating factor exhibits a complex effect on PMs, which is apparently associated both with the state of macrophages and with the structural features of the macrophage activator molecule.

In order to isolate the precursor from blood, we synthesized an affinity sorbent with cross-linked 25-hydroxy vitamin D_3_ and developed a chromatography procedure that differed completely from the method described previously in ref. [35]. 25-OH-D_3_/Sepharose^®^ turned out to be a stable substrate enabling up to nine chromatography runs and even more per portion of resin. Basically, the number of chromatographic isolation runs is limited only by resin stability. This study revealed unexpected properties of blood-derived DBP. The conducted cross-interaction affinity chromatography of DBP derived from the same donor (LEV) indicated that two different forms of DBP are isolated when using two different domains during affinity chromatography. If the 25-OH column is the primary one, only 10% of the macrophage-activating factor obtained from the initial amount of DBP eluted from the 25-OH-D_3_/Sepharose^®^ binds on the Actin column. If the Actin column is the primary one, less than 1% of the initial protein binds to the 25-OH-D_3_/Sepharose^®^ matrix. Since the molecular weight and presence of the antigenic Gc determinant of proteins obtained from the two columns are identical, it is fair to assume that the two types of DBPs differ in the tertiary structure of the polypeptide chain, which actually determines the MAF affinity [30]. An alternative explanation is related to the inhibition of specific affinity determinants in the protein molecule by low-molecular-weight derivatives of target ligands (actin or 25-OH-D_3_ degradants), which does not alter the apparent electrophoretic mobility of proteins. That can probably be the labile GcMAF-AlpI complex, which undergoes dissociation in the Laemmli system. Nonetheless, both variants activate PMs in an identical manner (data not presented), thus indicating that the glycosylation site is functionally active.

The result obtained means that when developing the therapeutic variant of the GcMAF preparation, one needs to choose the reliably standardizable method for isolating the macrophage-activating factor. The method of DBP production using 25-OH-D_3_/Sepharose^®^, which yielded the consistently isolatable two protein fractions, 58 kDa and 29 kDa, was selected for all our further studies.

Tandem mass spectrometry analysis identified the protein families to which both MAFs belonged. As expected, the protein with a molecular weight of 58 kDa was vitamin D3-binding protein; the protein with a molecular weight of 28 kDa was an ApoA1. Therefore, we have changed the terminology for the 58 kDa protein in further studies and started to use the name “GcMAF” (it was previously known as GcMAF-RF).

The experiments on direct interaction between two proteins during the sandwich-type blot assay of DBP/GcMAF and ApoA1 revealed that DBP did not bind to ApoA1, whereas direct molecular binding was observed for GcMAF (Figure 1(E2), shown with an arrow). This property has not been interpreted yet and needs to be further studied. The simultaneous isolation of two proteins on the 25-OH-D_3_/Sepharose^®^ column within a single complex can be attributed to the known functional binding between DBP (GcMAF) and ApoA1 through a vitamin D_3_ molecule. Both proteins occur within chylomicrons, which simultaneously contain ApoA1 and vitamin D_3_. Vitamin D_3_ is absorbed from the gastrointestinal tract, like other fat-soluble compounds. It is absorbed through the enterocyte and is released into the lymphatic system by getting incorporated in chylomicrons. In chylomicrons, vitamin D_3_ can bind to DBP through an affine site and be simultaneously associated with ApoA1 through the cholesterol residue [30,74,75,76]. These very complexes are presumably present in plasma. Two proteins can be isolated on the column either as a complex or individually. In the Laemmli system, proteins migrate in accordance with their molecular weights. ApoA1 is not glycosylated. Therefore, the results of deglycosylation-activating PMs refer to GcMAF only, while either ApoA1 does not participate in phagocyte activation at all or its involvement is minimal. As mentioned above, in our opinion, the more likely scenario is the binding of two blood plasma factors to two parts of the affinity matrix: the binding of ApoA1 and DBP to the cyclic cholesterol part and 25-OH-D_3_, respectively; in addition, these factors are most likely not part of a single complex.

The preparation obtained via affinity purification on a 25-OH-D_3_/Sepharose^®^ column stably consisted of the two fractions described above, had reproducible activating properties, and was used without separating the two factors.

### 3.2. Analysis of the Direction of Inflammatory Response Elicited in PMs Activated with GcMAF

In order to analyze the direction of the inflammatory response in PMs activated with GcMAF, we developed a procedure for phagocyte activation by a macrophage inducer and chose four major cytokines determining the direction of the inflammatory response. Pro-inflammatory mediators TNF-α and IL-1β, as well as anti-inflammatory cytokines inducing immune anergy or the development of T regulatory cells (TGF-β and IL-10), were selected. Leukocytes activated in different modes and purified enzymes were employed for conversion in this study. The conversion modes described earlier and in the Materials and Methods section were used. The strong focus placed on conversion with inflammatory leukocytes can be explained by the fact that it is inflammatory leukocytes that are involved in DBP conversion to GcMAF in the body [27,47]. We assessed the efficiency of mRNA synthesis of the specified cytokines using several GcMAF preparations depending on the dose and activation mode of leukocytes, which were used as a factor for converting the DBP precursor to the macrophage-activating factor GcMAF.

It was found that the unconverted precursor induces the mRNA synthesis of the specified cytokines in PMs with different efficiencies. Furthermore, DBP derived from different donors activates PMs toward synthesizing different cytokines. The results varied for different modes of conversion by leukocytes. After the conversion of DBP to GcMAF by leukocytes, the mode of synthesis of selected cytokines differs fundamentally: either pro- or anti-inflammatory cytokines, or simultaneously both pro- and anti-inflammatory ones, can be synthesized. It was found that in almost all the cases, both inflammatory response mechanisms are triggered in PMs.

The obtained result did not allow to identify the element of interaction that gives a clear idea of which direction of the inflammatory reaction in the phagocyte system will be induced.

A comparative analysis of the phagocytic activity of PMs and mRNA synthesis of the analyzed cytokines in PMs when using different doses of the macrophage-activating factor indicated that phagocytic activity as a normal response of phagocytes is found in both M1 (pro-inflammatory) and M2 (anti-inflammatory) PMs.

### 3.3. An Analysis of Direct Interaction between GcMAF and CLEC10A

An approach that we called sandwich-type blot assay has been elaborated; it allowed us to identify certain aspects of the GcMAF–PM interaction. It turned out that the well-known CLEC10A receptor residing on the plasma membrane of PMs directly binds to GcMAF under different engagement conditions. The interaction with both purified CLEC10A protein and PM lysate was revealed. The specific receptor/ligand engagement was found to be possible only in the presence of 20 mM Ca^2+^ ions in the Western blot buffer system and at least 2 mM Ca^2+^ ions in the culture medium of PMs. The three CLEC10A fractions interact with GcMAF: the 29 kDa protein and two high-molecular-weight derivatives (63 kDa and 65 kDa) [77,78,79]. The presence of Ca^2+^ ions affects the engagement of high-molecular-weight derivatives only. This approach allowed us to identify certain details of the interrelationships in the GcMAF/CLEC10A system. It turned out that the engagement of the macrophage-activating factor and the 65 kDa derivative is accompanied by the activation of the mRNA synthesis of pro-inflammatory cytokines. If interaction efficiency shifts towards receptor/ligand engagement with the 63 kDa derivative, mRNA synthesis in PMs switches to the synthesis of anti-inflammatory cytokines. In the conducted experiments, the interaction with two derivatives was modulating rather than absolute, which always causes certain ambiguity about the activation effect. Nonetheless, the findings were indicative of the involvement of two high-molecular-weight CLEC10A derivatives in activating the mRNA synthesis of both pro- and anti-inflammatory cytokines in PMs. It was demonstrated using the GcMAF produced by conversion with pure enzymes under the rapid conversion conditions that (1) the derivative predominantly interacting with GcMAF is the 63 kDa one and (2) this interaction inhibits the synthesis of pro-inflammatory cytokines and activates the synthesis of anti-inflammatory IL-10 (Figure 8E).

How can we explain these differences in cytokine synthesis activated by GcMAF preparations produced using different conversion modes when knowing that this protein molecule acts as an inducer?

The following factors are involved in activation events: an inducer (DBP, GcMAF), CLEC10A receptor, and, in particular, the 29 kDa canonical protein that binds both to DBP and GcMAF regardless of whether the terminal or partially deglycosylated GalNAc is present or absent. High-molecular-weight derivatives (~63 kDa, 65 kDa), a concentration of Ca^2+^ ions that causes epitope opening in the high-molecular-weight carbohydrate-binding CLEC10A derivative, and PM cells are also present. It is suggested that the direction of inflammatory activity of PMs is related to the structural variants of the glycosylation site, its location in the macrophage-activating factor whose status determines whether pro- or anti-inflammatory cytokines will be synthesized, and possibly the number of receptor–ligand interactions (Figure 9). The presence of several glycosylation sites (Tre^18^/Tre^20^) and, at the same time, a mixture of molecules with different depths of deglycosylation creates an ambiguous picture of the activation of PMF mRNA synthesis, which significantly complicates the assessment of the internal logic of events.

Two types of GcMAF (DBP)/CLEC10A binding are detected experimentally. DBP binds to the canonical 29 kDa CLEC10A receptor via the bond not requiring the activation of the GalNAc-specific epitope by Ca^2+^ ions. Upon this binding, the receptor induces the minor chaotic synthesis of the analyzed cytokines through an unknown mechanism. Ostensibly, the signaling pathway of mRNA synthesis induction is independent of the intracellular concentration of calcium ions in this case. The emergence of Ca^2+^ ions induces a different specific binding between GcMAF and CLEC10A through two high-molecular-weight receptor derivatives that are present in the purified commercial protein preparation (and assumedly, in PM lysate) and is responsible for the direction of mRNA synthesis.

These results explain the differences in mRNA synthesis in PMs activated by different GcMAF preparations as follows. Two high-molecular-weight CLEC10A derivatives are responsible for the direction of inflammatory response in PMs: the 65 kDa derivative is responsible for the pro-inflammatory response while the 63 kDa derivative is responsible for the anti-inflammatory one. Both epitopes are calcium-dependent. The 29 kDa CLEC10A epitope apparently acts as a primary “anchor” for GcMAF and is responsible for its orientation in a clathrin-coated caveola [80,81,82].

An important outcome of this study is the finding that the engagement with Ca^2+^-dependent receptor derivative only is insufficient for eliciting response in PMs as it follows from the absence of the effect of activating PMs with pure GalNAc. Supposedly, the primary orienting interaction of the GcMAF molecule with the 29 kDa CLEC10A is required. The stabilization of this binding in the presence of Ca^2+^ ions is followed by the engagement of the glycosylation site of the GcMAF molecule with high-molecular-weight CLEC10A derivatives residing in immediate proximity to the 29 kDa CLEC10A. The choice of the allelic form of the 65 kDa or 63 kDa protein is assumed to depend on the depth of GcMAF deglycosylation at the glycosylation site(s).

If all the three allelic forms of the CLEC10A receptor interact with GcMAF of various deglycosylation degrees, the simultaneous interaction with both allelic protein forms is possible, which will be accompanied by the activation of synthesis of both pro- and anti-inflammatory cytokines as it follows from the findings.

### 3.4. Aggregation of CLEC10A Receptor Molecules as a Mechanism Inhibiting mRNA Synthesis of TNF-α, IL-1β, TGF-β, and IL-10 in PMs

In some variants, mRNA synthesis of anti-inflammatory cytokines, both in the ex vivo and in vivo experiments, follows a clearly defined a non-monotonic (bell-shaped) pattern. The efficiency of mRNA synthesis is insignificant but clearly depends on the dose of the macrophage-activating factor GcMAF: it increases at doses of 0.02 µg or 0.2 µg and is completely inhibited at the dose of 2.0 µg. The mRNA synthesis of pro-inflammatory cytokines simultaneously behaves in an identical or nearly identical manner.

The mechanism of non-monotonic (bell-shaped) response to the dose has been thoroughly studied when inducing the TCR-mediated inflammatory response and is related to the internalization of Ca^2+^ ions associated with specific receptors by the cell. For the studied system, the receptor can be internalized only when the following complex is formed (for CLEC10A): receptor + three Ca^2+^ molecules + ligand on any substrate [83,84]. A clathrin vesicle (caveola) with the active receptor is immersed in the cytoplasm and fuses with an endosome, where Ca^2+^ is released; subsequently it is released into the cytosol due to activated IP3/IP3R pumping. The amount (local concentration) of cytosolic Ca^2+^ ions is the key aspect responsible for the direction of cellular response. At a very low concentration of Ca^2+^ ions, calcineurin phosphatase is activated and brings about conditions for the emergence of the specific NAFT-AP-1 complex on which the activation of pro-inflammatory genes depends. When the ligand and receptor concentrations are equimolar, an excessive number of Ca^2+^ ions infiltrate the endosomes and the cytosol, thus causing calcineurin hyperactivation and the functional degradation of protein kinanse II and Pyk2 on which AP-1 activation depends. When AP-1 is inactive, NAFT activates the anti-inflammatory cytokine genes and IL-10 in particular [85,86,87,88,89,90]. In other words, in the case of CLEC10A, all the available CLEC10A receptors will be activated, the cytosol will be saturated with Ca^2+^ ions, and the suppressive cellular phenotype will be formed at an equimolar amount of the ligand carrying free terminal GalNAc moiety [60,83,91].

We have probably detected this very form of response after treating PMs with the GcMAF preparation produced by enzymatic conversion in a solution. We found that the mRNA synthesis of pro-inflammatory cytokines TNF-α and IL-1β is completely inhibited in this variant while IL-10 synthesis is activated. Specific receptor/ligand engagement with the 63 kDa band is clearly detected.

Nevertheless, this study offers examples of a different type of response (when pro- and anti-inflammatory cytokines are synthesized simultaneously). Experimental studies are needed for understanding the mechanism of this activation of PMs.

As mentioned above, the dose of the ligand-carrying carbohydrate that binds to the CLEC10A receptor accepting three Ca^2+^ molecules affects the direction of mRNA synthesis in PMs. Supposedly, receptor aggregation takes place as the dose is increased, and cytokine mRNA synthesis is inhibited. The following mechanism for such interaction has been proposed in the literature analyzed. Upon stochastic aggregate formation, CD45 phosphatase, which is ubiquitously expressed in antigen-presenting cells and has an affinity for CLEC10A, is integrated into it [91,92,93]. This factor dephosphorylates the cytoplasmic domains of “bundled” receptor molecules, thus blocking their internalization and completely stopping the entry of Ca^2+^ ions into the cell. Therefore, any synthesis processes are inhibited. This very situation is observed in the experiments focusing on the activation of cytokine mRNA synthesis with 2 µg “C GcMAF LEV L pure 37 °C”. Furthermore, the conducted cytological experiments revealed that at a dose of 2 µg, numerous cells with large fluorescent aggregates are detected on the plasma membrane. Meanwhile, at lower doses, the label is detected in different cellular compartments. These findings are consistent with the literature data [53] demonstrating that the cell surface receptors can be aggregated, which is accompanied by the loss of their specific functions.

## 4. Materials and Methods

### 4.1. Experimental Animals

We used male and female 2- to 6-month-old C57BL/6 mice (weight, 18–24 g) bred at the Common Use Center Vivarium for Conventional Animals of the Institute of Cytology and Genetics of the Siberian Branch of the Russian Academy of Sciences (Novosibirsk, Russia). The animals were kept in groups of 6–10 mice per cage with free access to food and water. The animals were sacrificed using the method of cervical dislocation. All the experiments involving animals were conducted in strict compliance with the principles of humanity in accordance with the European Community Council Directives (86/609/EEC) and were approved by the Animal Care and Use Committee of the Institute of Cytology and Genetics SB RAS.

### 4.2. GcMAF Preparation

Vitamin D_3_-binding protein (DBP) was isolated from human plasma collected from individual donors using either affinity chromatography on a 25-OH-D_3_/Sepharose^®^ or, in specified cases, via actin–sepharose or actin–chitin affinity chromatography [36,37]. DBP derived from an individual donor was denoted with the respective letters or a number (e.g., DBP LEV means that DBP was obtained from donor LEV). Plasma derived from other donors was denoted with numbers 1–3. In our study, the resulting DBP was converted to GcMAF using several methods.

#### 4.2.1. The First Method for GcMAF Production Is Denoted as Conventional (C)

To produce GcMAF, DBP immobilized on 25-OH-D_3_/Sepharose^®^ was eluted with 3M guanidine chloride, dialyzed against PBS, and incubated in the presence of leukocytes preliminarily purified to remove red blood cells (L pure). Red blood cell lysis was performed using a buffer (0.15 M NH_4_Cl, 0.01 M KHCO_3_, and 0.1 mM EDTA). DBP was incubated with purified leukocytes in the solution supplemented with 10% FBS and CaCl_2_ at a final concentration of 1.5 mM in an atmosphere of CO_2_ at 37 °C during 24 h. After the conversion under conventional conditions, the GcMAF preparation was sterilized via filtering. GcMAF preparations produced under the conventional conditions are denoted as follows: “C GcMAF L pure 37 °C” means that the preparation was produced under the conventional conditions using purified activated leukocytes; incubation was carried out at 37 °C. “C GcMAF L plasma 39 °C” means that the preparation was produced under the conventional conditions using leukocytes pre-activated in native plasma at 39 °C (L plasma 39 °C) for 2 h.

#### 4.2.2. The Second Method for GcMAF Production Is Denoted as the Rapid (Express) Method (Ex)

Conversion was carried out when the precursor was bound to the substrate on resin. Leukocytes treated using different methods—purified non-activated leukocytes (L pure), leukocytes pre-activated with lysophosphatidylcholine (LysoPC, Sigma, St. Louis, MO, USA) (L LysoPC), leukocytes pre-activated in native plasma at elevated temperature of 39 °C (L plasma 39 °C), and leukocytes pre-activated in plasma at 37 °C (L plasma 37 °C)—were added to the conversion medium containing resin associated with DBP. Activation of leukocytes purified using LysoPC or temperature of 39 °C was carried out in buffer A (2 mM Tris HCl of pH = 8.0, 0.5 mM β-mercaptoethanol, 0.2 mM CaCl_2_). GcMAF preparations produced using this rapid (express) method were denoted as follows. The denotation “Ex GcMAF L plasma 37 °C” indicates that the preparation was produced using the rapid method (namely, by adding leukocytes activated in plasma at 37 °C for 3 h to DBP immobilized in the column); the denotation “Ex GcMAF L plasma 39 °C” means that the preparation was produced using the rapid method by adding leukocytes pre-activated in native plasma at 39 °C for 3 h to DBP immobilized in the column and their co-incubation at 37 °C for 20 h in the presence of 1.5 mM CaCl_2_ occurred; “Ex GcMAF L plasma LysoPC 37 °C” denotes that the preparation was produced using the rapid method by adding leukocytes pre-activated with LysoPC in native plasma at 37 °C for 3 h to DBP immobilized in the column and their co-incubation at 37 °C for 20 h in the presence of 1.5 mM CaCl_2_ occurred. “Ex GcMAF L pure LysoPC 37 °C” means that the preparation was produced using the rapid method by adding pre-purified leukocytes and then activated by adding LysoPC at 37 °C for 3 h to the DBP immobilized in the column and their co-incubation at 37 °C for 20 h in presence of 1.5 mM CaCl_2_ occurred.

The GcMAF preparations produced by the aforementioned treatments were washed to remove the conversion medium after incubation for 20 h, eluted with 3M guanidine chloride, dialyzed against PBS, and used for activating peritoneal macrophages (PMs). Digits and letters specifying the origin of GcMAF indicate the donors of plasma used to obtain DBP, the GcMAF precursor. These preparations are owned by open joint-stock company “ACTIVATOR MAF”.

Analysis of the effect of leukocyte products used for GcMAF production on PM ability to express the analyzed cytokines.

In order to assess the effect of leukocyte products used to obtain GcMAF, we modified the conditions for producing the latter (leukocyte-conditioned medium) by excluding DBP. To test the possibility of PMF activation by the conditioned leukocyte medium, we added only leukocyte medium to the wells in amounts similar to those obtained when producing GcMAF and in the same volume as for the activator dose dilution. We added 2 μg of “C GcMAF L pure” in 30 μL of conversion leukocyte medium per well for PM activation and 30 μL of conversion medium lacking the GcMAF activator obtained under similar culture conditions for assessment of the effect of pure leukocyte medium. For comparison, the medium was titrated 10- and 100-fold to 3 and 0.3 µL per well, respectively (Figure 2A).

### 4.3. Ex Vivo Activation of PMs with DBP, GcMAF, and ApoA1

PMs were isolated from intact C57BL/6 mice and precipitated using centrifugation at 400× *g* for 7 min. The cells were resuspended in RPMI; concentration was counted in the Goryaev’s chamber. PMs (1 × 10^6^ cells/well) were cultured in RPMI-1640 medium (BioloT, St. Petersburg, Russia) supplemented with 10% FBS (HyClone, Logan, UT, USA) and 40 μg/mL gentamicin in 24-well plates for 12 h. Next, the medium was changed to RPMI-1640 in the absence (control) or presence of DBP, GcMAF, or ApoA1. The macrophage-activating factors were added to each well (four replicates per dose): 0.02 µg, 0.2 µg, 2 µg, and 20 µg. The cells were incubated in an atmosphere of CO_2_ (Memmert, USA LLC, Eagle, WI, USA) at 37 °C for 3 h. After 3 h, the cells were washed to remove the MAF. The activated and control PMs were lysed with TRIzol Reagent (Thermo Fisher Scientific, Waltham, MA, USA) to obtain total RNA. Additionally, to assess the effect of ApoA1 on the synthesis of mRNA isolated from PMs, the initial DBP preparation was ultra-fractionated by concentrator MWCO 30,000 (Amicon, Burlington, MA, USA). Purified ApoA1 was used to activate PMs at a dose of 2 µg (Appendix A). PM activation by GcMAF was conducted in non-supplemented RPMI-1640 medium. Real-time PCR results were normalized to the level of mRNA isolated from PMs treated only with non-supplemented RPMI-1640. After activation, all samples were incubated under absolutely identical conditions, and the effect observed was caused solely by GcMAF.

### 4.4. Ex Vivo Activation of J774 Cells with GcMAF

Macrophage J774 cell line was received from the cell collection of the Institute of Cytology and Genetics SB RAS. For activating the J774 cells with GcMAF, the cells were cultured in DMEM medium (BioloT, St. Petersburg, Russia) supplemented with 10% FBS (HyClone, Logan, UT, USA) and 40 μg/mL gentamicin in 24-well plates (1 × 10^6^ cells/well) at 37 °C in a 5% CO_2_ incubator (Memmert, USA LLC, Eagle, WI, USA) to 70–80% confluency for 12 h. Next, the activation procedure was performed according to the scheme identical to that for ex vivo activation of PMs with GcMAF.

### 4.5. In Vivo Activation of Peritoneal Macrophages with GcMAF

To perform in vivo activation of PMs with GcMAF, we took three C57BL/6 mice in each of the four study groups and one control group. Mice received intravenously a single dose of GcMAF (0.02 µg, 0.2 µg, 2 µg, and 10 µg) in 200 µL of saline. Mice in the control group were injected with saline only. Five hours post-injection, PMs were isolated in RPMI-1640 medium individually for each mouse, precipitated, and lysed with TRIzol Reagent (Thermo Fisher Scientific, Waltham, MA, USA) to obtain total RNA.

### 4.6. Analysis of Activation of Phagocytic Activity of Peritoneal Macrophages in Mice

We used streptavidin-coated beads in our studies. Two mechanisms of target determination and phagocytosis are possible. In the first case, cells detect streptavidin as a PAMP, which is the uptake trigger. Primary “opsonization” of particles by intercellular matrix factors secreted by phagocytes followed by receptor-mediated phagocytosis is possible. In cases when phagocytosis objects are not living cells but pieces of coal, asbestos, glass, metal, etc., phagocytes are known to first make the object of absorption acceptable for the reaction by enveloping it with their own products, including intercellular matrix components produced by them [94,95]. The phagocytic function of macrophages was assessed using the technique presented in [37,94].

### 4.7. Mass Spectrophotometry Analysis of Vitamin D3-Binding Protein

The data on peptide structure were obtained via tandem mass spectrometry analysis that involved separation of precursor ions, fragmentation of these ions yielding secondary ions, and analysis of fragment ions. To perform mass spectrometry analysis of the samples pre-separated by 1D electrophoresis, sample components were enzymatically cleaved with trypsin; impurities were removed, and the resulting peptide mixtures were subsequently analyzed individually. For this purpose, selected bands were cut out of the gel, placed into 1.5 mL polypropylene tubes, washed with water (twice) and 50% acetonitrile (Merck, Darmstadt, Germany) in 0.2 M ammonium bicarbonate (Sigma-Aldrich, Burlington, MA, USA), and dried in 100% acetonitrile (Merck, Darmstadt, Germany). For reduction, we added 2 µL of 20 mM dithiothreitol (BioRad, Hercules, CA, USA) in a buffer containing 0.2 M ammonium bicarbonate (Sigma-Aldrich, Burlington, MA, USA), and the tubes were incubated in a TS-100 thermoshaker (Biosan, Riga, Latvia) for 30 min at 600 rpm and 57 °C. For alkylation, we added 2 µL of 50 mM iodoacetamide (BioRad, Hercules, CA, USA) in the buffer containing 0.2 M ammonium bicarbonate (Sigma-Aldrich, Burlington, MA, USA), and the tubes were incubated in dark for 30 min in the TS-100 thermoshaker (Biosan, Riga, Latvia) at 600 rpm. The contents of the tube were washed and dried in 100% acetonitrile (Merck, Darmstadt, Germany). To conduct tryptic hydrolysis of the protein, we added 0.02 mM trypsin (Trypsin Gold, Mass Spectrometry Grade, Promega, Madison, WI, USA) in 50 mM ammonium carbonate (Sigma-Aldrich, Burlington, MA, USA) and performed incubation in the TS-100 thermoshaker (Biosan, Riga, Latvia) for 14 h at 600 rpm and 37 °C. Peptide extraction was performed using 1% trifluoroacetic acid (Sigma-Aldrich, Burlington, MA, USA). Peptides were purified on a Millipore ZIPTIP C18 column (Merck, Darmstadt, Germany) according to the manufacturer’s protocol. Peptide mixtures were dried and redissolved in 20 µL of 0.1% trifluoroacetic acid solution (Sigma-Aldrich, Burlington, MA, USA) in an acetonitrile (Merck, Darmstadt, Germany)/water (W6-1, Fisher Chemical, Waltham, MA, USA) mixture (2/98% *v*/*v*). A Thermo Fisher Scientific Ultimate 3000 Series HPLC system (Nano/Cap System NCS-3500RS), a WPS-3000 TPL RS autosampler (with a 20 µm sample loop), and a sample cooling system were used for chromatographic fractionation. The fractionation of compounds was performed on an Acclaim PepMap RSLC C18 column (75 µm × 150 mm, 2 µm, 100 Å) in the gradient mode. The composition of the mobile phase (the eluent) was as follows. Component A: 0.1% formic acid solution (Thermo Scientific, Waltham, MA, USA) in water (W6-1, Fisher Chemical, Waltham, MA, USA). Component B: 0.1% formic acid solution (Thermo Scientific, Waltham, MA, USA) in an acetonitrile (Merck, Darmstadt, Germany)/water (W6-1, Fisher Chemical) mixture (80:20% *v*/*v*). Load pump mobile phase: 0.1% trifluoroacetic acid (Sigma-Aldrich, Burlington, MA, USA) in acetonitrile (Merk, Darmstadt, Germany)/water (W6-1, Fisher Chemical, Waltham, MA, USA) (2%/98% *v*/*v*).

The compounds were identified on an Orbitrap Fusion Lumos mass spectrometer under the following conditions: electrostatic spray ionization was performed at atmospheric pressure, and an orbitrap was used as a sensor (scanning modes: MS OT, ddMS^2^ OT HCD; resolution, 60,000; high-energy collision-activated dissociation cell; dissociation energy, 30%; detectable charge state, 2–7; a quadrupole mass filter used for precursor isolation; mass isolation window width, 1.6 m/z; detectable mass range, 300–1500 m/z; detection of positively charged ions; intensity threshold, 5 × 104; ion accumulation time, 60 ms; number of scans acquired, 1; dynamic exclusion duration, 60 s; sprayer voltage, 2.0 kV; capillary temperature, 305 °C; and sprayer temperature, 270 °C). The Xcalibur v.2.4 software was used for collecting and processing the mass spectrometry data.

The mass spectrometry data were uploaded as raw files into the Proteome Discoverer (2.4.0) software. The data were processed using the standard processing step involving the following elements: Spectrum files, Spectrum selector, SequestHT, Percolator, and IMP-ptmRS; the consensus step involved the elements MSF Files, PSM Grouper, Peptide Validator, Peptide and Protein Filter with a link to protein annotation, Protein Score with a link to Protein FDR Validator, and Protein Grouping. The data on the amino acid sequence obtained by analyzing the MS1 and MS2 mass spectra were compared to the amino acid sequence of human genes from the UP000005640 proteome.

### 4.8. Obtaining cDNA

Total RNA from peritoneal macrophages was isolated using TRIzol Reagent (Thermo Fisher Scientific, Waltham, MA, USA) following the manufacturer’s instructions. The amount of RNA was measured on a Qubit 4 fluorometer (Thermo Fisher Scientific, Waltham, MA, USA). The reverse transcription PCR was carried out on a poly-A mRNA template using a T100 Thermal Cycler amplifier (Bio-Rad Laboratories, Inc., Hercules, CA, USA) and an MMLV RT kit (Evrogen, Moscow, Russia) according to the manufacturer’s protocol.

### 4.9. PCR Analysis

PCR primers for coding regions of each pro-inflammatory and anti-inflammatory cytokine gene were constructed using the Vector NTI v.9 software (Life Technologies, Wilmington, DE, USA) and synthesized by BIOSSET Ltd. (BIOSSET, Novosibirsk, Russia). The sequences of primers used in this study are listed in Table 1.

PCR was carried out in a total volume of 50 µL; the reaction mixture contained Taq-buffer, 2 mM MgCl_2_, 0.2 mM dNTP, 0.2 pmol of forward and reverse primers, 1 ng of the template, and 5 units of Taq-polymerase (the reagents for PCR were provided by Medigen, Novosibirsk, Russia). The PCR scheme was as follows: cycle 1 (×1): 95 °C—3 min; cycle 2 (×33): 95 °C—30 s; 59 °C—30 s; 72 °C—40 s; cycle 3 (×1): 72 °C—5 min; storage at 10 °C.

### 4.10. Real-Time PCR

Real-time PCR was carried out in 96-well plates using BioMaster HS-qPCR SYBR (2x) (BIOLABMIX LLC, Novosibirsk, Russia) according to the manufacturer’s protocol on a QuantStudio5 PCR system (Thermo Fisher Scientific, Waltham, MA, USA). Real-time qPCR analysis of each sample was performed in three replicates. The relative expression level was determined using the 2^–ΔΔCt^ method. Intact nontreated PMs were used as the control group; the expression level of the target gene in them was assumed to be equal to 1. The GAPDH gene was used as reference. The cycling parameters were as follows: 95 °C for 10 min, 40 cycles of 95 °C for 30 s, 59 °C for 30 s, and 72 °C for 30 s, with a final melting step involving slow heating from 6 to 95 °C.

### 4.11. Interaction of GcMAF and Pure GalNAc with Anti-Gc Antibodies

In order to compare the interaction of GcMAF and pure galactosamine (GalNAc) with anti-Gc antibodies, samples of the compounds were applied onto a nitrocellulose membrane: GalNAc (Sigma-Aldrich) at an equimolar amount and at 10-fold and 100-fold excess with respect to the amount of GcMAF. GalNAc (at the amounts specified above) in 110 µL of 0.01 M PBS supplemented with 0.02% Tween 20 was placed into wells of a vacuum dot blotting system. GcMAF (20 µL, 4.5 µg) was placed into the fourth well of the system. The setup was connected to vacuum pump; after the contents of the wells had passed through the membrane, we conducted a successive series of fixation treatments (vacuum infiltration) of the membrane with methanol (100 µL per treatment) and three washing procedures using 50 µL of transfer buffer per washing. Once pumping had been completed, the membrane was cut into two parts: the first part contained GalNAc applied three times, while the other part contained GcMAF. The membranes were treated with blocking buffer (0.01 M PBS supplemented with 0.02% Tween 20) for 1.5 h. Anti-Gc antibodies (Cloud-Clone-Corp, Houston, TX, USA) were then added, and overnight incubation on a rocker at +9 °C was performed.

### 4.12. Western Blot Assay of the Lysate of Macrophages with Anti-CLEC10A Antibodies

Western blot assay was conducted to prove that the macrophage surface contained CLEC10A. Freshly isolated PMs (10^7^) were lysed in 100 µL of saline containing 15 µL of 0.1 M PMFS and 40 µL of Sample Buffer (0.066 M Tris-HCl pH = 6.8; 26.3% glycerol; 2.1% SDS; and 0.011% Bromophenol Blue) for 10 min, boiled for 10 min, and centrifuged. Lysate (2 × 10^6^ PM cells per lane) was applied to perform electrophoresis. The commercially available CLEC10A (Cloud-Clone-Corp, Houston, TX, USA) (2 µg per lane) was used as control. Western blot assay with antibodies was conducted after electrophoresis followed by electroblotting onto a NC membrane. Nonspecific binding was blocked by incubation in 0.01 M phosphate-buffered saline (PBS) containing 0.02% Tween 20 overnight at 4 °C. For Western blot, the blots were probed with anti-CLEC10A antibodies (Cloud-Clone-Corp, Houston, TX, USA). The blots were developed using ECL Western blotting detection system (Abcam, Cambridge, UK) and imaged on an iBright imager (Thermo Fisher Scientific, Waltham, MA, USA).

### 4.13. An Analysis of Protein–Protein Interactions between GcMAF and CLEC10A (Sandwich-Type Assay)

The direct GcMAF–CLEC10A interaction was studied using the sandwich-type assay; after electrophoresis, CLEC10A (Cloud-Clone-Corp, Houston, TX, USA) or GcMAF was transferred from polyacrylamide gel to the NC membrane using the procedure described above and treated with a counteragent. The weight of proteins applied onto the gel was 2 µg; 15 µg of the counteragent in 10 mL of 0.01 M PBS supplemented with 0.02% Tween 20 and 20 mM CaCl_2_ was used for treating the membrane. After overnight incubation, the membranes were washed thrice with a buffer (0.01 M PBS supplemented with 0.1% Tween 20), with each washing procedure lasting 5 min, and incubated in the presence of 5 µg of anti-Gc (Cloud-Clone-Corp, Houston, TX, USA) or anti-CLEC10A antibody (Cloud-Clone-Corp, Houston, TX, USA) in 10 mL of the buffer consisting of 0.01 M PBS supplemented with 0.02% Tween 20 for 10 h. Antibody response was detected after washing using ECL Western blotting detection system (Abcam) and imaged on an iBright imager (Thermo Fisher Scientific, Waltham, MA, USA).

### 4.14. Fluorescent Cytology Analysis of Interaction between PMs and Cy5-GcMAF

GcMAF was labeled using sulfo-Cyanine5 (Cy5) antibody labeling kit (Lumiprobe, USA) according to the manufacturer’s protocol. Freshly isolated PMs (1 × 10^6^ cells/well) were cultured in 500 µL of RPMI-1640 (BioloT, St. Petersburg, Russia) supplemented with 10% FBS (HyClone) and 40 µg/mL gentamycin in 24-well plates (Eppendorf SE, Germany) for 12 h. Next, 2.5 µg/mL Cy5-GcMAF was added to each well. The plate was analyzed using an LSM 780 NLO confocal microscope (Carl Zeiss AG, Germany). Fluorescence intensity of the probe inside the cell was analyzed using the ZEN software (Carl Zeiss AG, Germany).

### 4.15. Statistical Analysis

Statistical analysis was performed using the Statistica 10 software (StatSoft, Tulsa, OK, USA). The graphs were designed either using Microsoft Excel 2013 (Microsoft, Redmond, Washington, DC, USA), Statistica 10 (StatSoft, Tulsa, OK, USA), or GraphPad Prism9.3.1 (Graph-Pad Software, San Diego, CA, USA) software. The graphs show Means ± Standard Deviations (SDs). The validity of differences was evaluated using the Mann–Whitney U test. The revealed differences were considered statistically significant at *p* < 0.05 (Mann–Whitney U test).

## 5. Conclusions

DBP binds to the 29 kDa CLEC10A but does not bind to the 63 kDa or 65 kDa CLEC10A. Like DBP, GcMAF binds to the 29 kDa CLEC10A and also to two CLEC10A derivatives (63 kDa and 65 kDa) in the presence of 20 mM Ca^2+^ ions. GcMAF can have different variants of both glycosylation and the location of the glycosylation site. The results of this study do not allow us to determine exactly which variant of glycosylation is responsible for the pro- or anti-inflammatory response vector of PMF. Nevertheless, it can be said that as a result of various treatments, two principal variants of the GcMAF molecule can be formed: binding 65 kDa and 63 kDa derivatives of CLEC10A. It is this form of interaction that determines the direction of the inflammatory response of PMF. Furthermore, the conducted analysis allows one to see the aggregation mechanism of the receptors anchored in clathrin vesicles (caveolae) that looks as follows (Figure 9).

With the excess of the ligand, competitive interactions between the “anchoring” 29 kDa derivative and high-molecular-weight derivatives 63 kDa and 65 kDa emerge in the presence of Ca^2+^ ions. It is believed that the aggregation of complexes takes place when the “anchoring” 29 kDa/63 kDa/65 kDa complex cannot be formed and simultaneously the receptor/ligand engagement of the high-molecular-weight Ca^2+^-dependent CLEC10A derivatives and GcMAF molecules excessively present in the cellular environment occurs. Tails of a molecule of different clathrin vesicles (caveolae) can interact with each other to form clathrin–clathrin (intercaveolar) intermolecular bridges, as it has been reported for collagen upon the formation of ordered intercellular scaffold structures [96,97,98]. Clathrin vesicles (caveolae) are constricted and aggregates are formed that are observed when analyzing PMs treated with 2 µg GcMAF. The synthesis of both pro- and anti-inflammatory cytokines is inhibited when there are no conditions for functional complex formation and its internalization by the cytoplasm.

An important remark in relation to the explanation for the GcMAF action provided above is that its molecule carries an actin/vitamin D_3_ binding site. These additional active GcMAF components can have a significant effect on the mechanism of PM activation described above.

Therefore, the direction of cellular response is determined by the direction of signaling, which is a quantitative parameter and depends on the interaction between the CLEC10A receptor residing on macrophages and dendritic cells and the glycosylation site of GcMAF.

## 6. Patents

The production of the GcMAF preparation is patented: priority No. 047390 2023121663 from 17 August 2023.

## Figures and Tables

**Figure 1 ijms-24-17396-f001:**
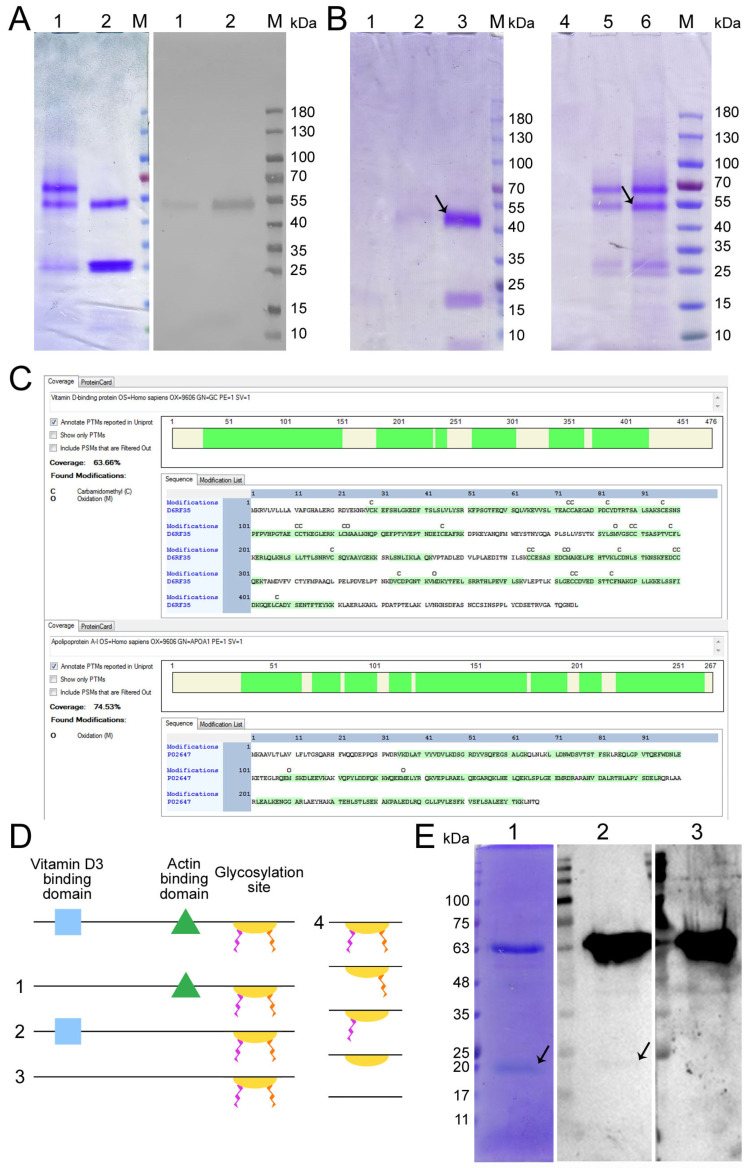
The structural and functional characterization of DBP. (**A**) Polyacrylamide gel of DBP samples obtained using actin–chitin resin (1) and 25-OH-D_3_/Sepharose^®^ (2) affinity chromatography (left) and Western blot of these samples with anti-Gc antibodies (right). (**B**) Polyacrylamide gels of DBP samples obtained via sequential dual-affinity chromatography: 25-OH-D_3_/Sepharose^®^/actin–chitin chromatography (1–3) and actin–chitin/25-OH-D_3_/Sepharose^®^ chromatography (4–6), with breakthrough after the second chromatography step (1 and 5), DBP samples obtained using sequential dual-affinity chromatography (2 and 4), and the initial DBP obtained via the first chromatography step, arrows denote DBP (3 and 6). M—molecular marker “The Thermo Scientific™ Page Ruler™ Prestained protein Ladder” (Thermo Fisher Scientific Inc., Carlsbad, CA, USA). (**C**) The result of comparing the amino acid sequences of the two bands at 58 kDa and 28–30 kDa with those from the UP000005640 proteome. (**D**) A schematic view of the DBP structure and presumed variants of the deglycosylation of GalNAc-sialic acid-galactose trisaccharide. The two “tails” represent sialic acid and galactose. No “tails” denote only GalNAc. (**E**) Polyacrylamide gel of DBP3 samples obtained using 25-OH-D_3_/Sepharose^®^ affinity chromatography (1) and sandwich-type Western blot of (DBP3//GcMAF3 Ex L pl 39 °C/αGc) (2) and (DBP3//αGc) (3). The arrows show apolipoprotein A1 A1 (ApoA1).

**Figure 2 ijms-24-17396-f002:**
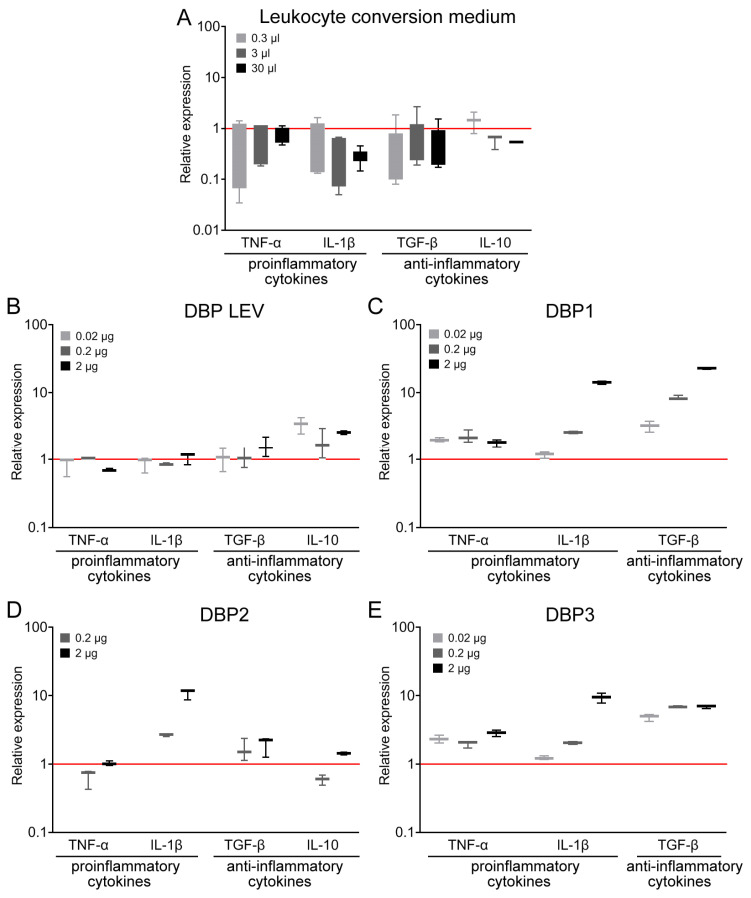
Quantification of mRNA expression of the cytokine genes in peritoneal macrophages of mice treated with leukocyte conversion media and DBP derived from different donors. (**A**) The diagram shows mRNA expression of the cytokine genes in peritoneal macrophages of mice treated with leukocyte conversion medium. Doses correspond to 0.3 µL, 3 µL, and 30 µL of leukocyte conversion medium obtained under similar leukocyte culture conditions but without GcMAF (see Materials and Methods for more details). The control here comprises untreated peritoneal macrophages, whose expression level is considered 1 (the red solid line). (**B**–**E**) mRNA expression of cytokine genes in peritoneal macrophages of mice treated with DBP derived from different donors with 0.02 µg, 0.2 µg, and 2 µg (0.2 µg, and 2 µg for **D**) compared to the control (untreated), whose expression level is considered 1 (the red solid line). (**B**) DBP LEV. (**C**) DBP1. (**D**) DBP2. (**E**) DBP3. The median values, the interquartile range, and the minimum and maximum values are provided. IL-10 is not shown in (**C**,**E**) due to undetectable level of corresponding mRNA in these samples.

**Figure 3 ijms-24-17396-f003:**
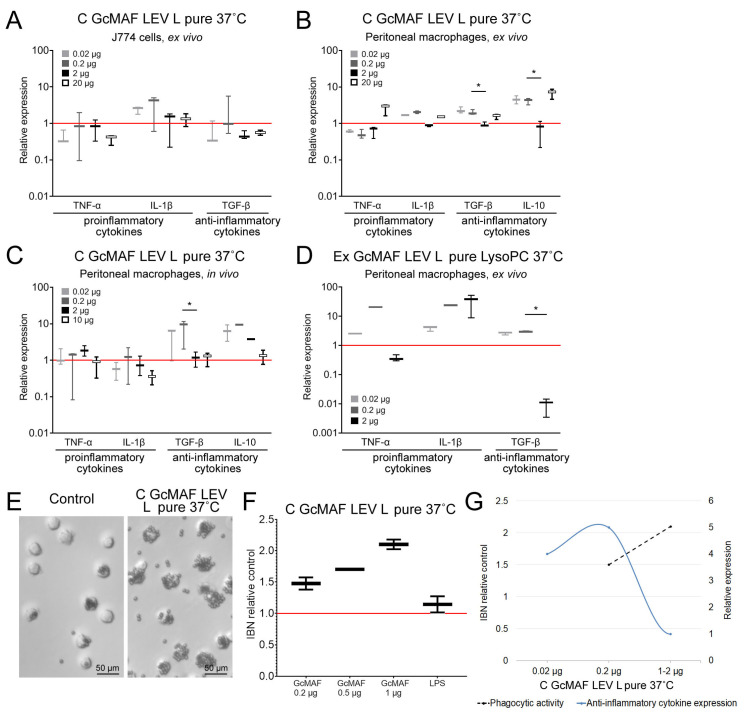
The effects of different GcMAF specimens derived from one donor (LEV) on cytokine expression and by J774 macrophage cell line and PMs ex vivo and in vivo and phagocytic activity of murine PMs with 0.02 µg, 0.2 µg, 0.5 µg, 1 µg, 2 µg and 10–20 µg (with the exception of (**D**) with 0.02 µg, 0.2 µg, and 2 µg) preparation with respect to the control (untreated), whose expression level was assumed to be 1 (the red solid line). (**A**–**C**) mRNA expression of the cytokine genes in J774 cells and in peritoneal macrophages of mice treated with C GcMAF LEV L pure 37 °C. (**A**) J774 cells, ex vivo. (**B**) Peritoneal macrophages, ex vivo. (**C**) Peritoneal macrophages, in vivo. (**D**) mRNA expression of the cytokine genes in peritoneal macrophages of mice ex vivo treated with Ex GcMAF LEV L pure LysoPC 37 °C. (**E**,**F**) Assessment of the phagocytic activity of peritoneal macrophages treated with C GcMAF LEV L pure 37 °C according to their ability to internalize metallic beads. (**E**) Images of the beads phagocytized by naïve macrophages (control) and macrophages treated with C GcMAF LEV L pure 37 °C. (**F**) Quantification of the phagocytic activity of PMs (IBN) treated with C GcMAF LEV L pure 37 °C and LPS (10 µg/mL) as a positive control. The naïve macrophages were used as a control; their phagocytic activity was assumed to be 1 (the red solid line). (**G**) Comparison of the phagocytic activity (dashed line) and mRNA expression of anti-inflammatory cytokine genes (non-monotonic) in murine PMs activated with different doses of C GcMAF LEV L pure 37 °C. IL-10 is not shown in (**A**,**D**) due to undetectable level of corresponding mRNA in these samples. We noted significance of differences between GcMAF doses of 0.2 µg and 2 µg (* *p* < 0.05; Mann–Whitney U test).

**Figure 4 ijms-24-17396-f004:**
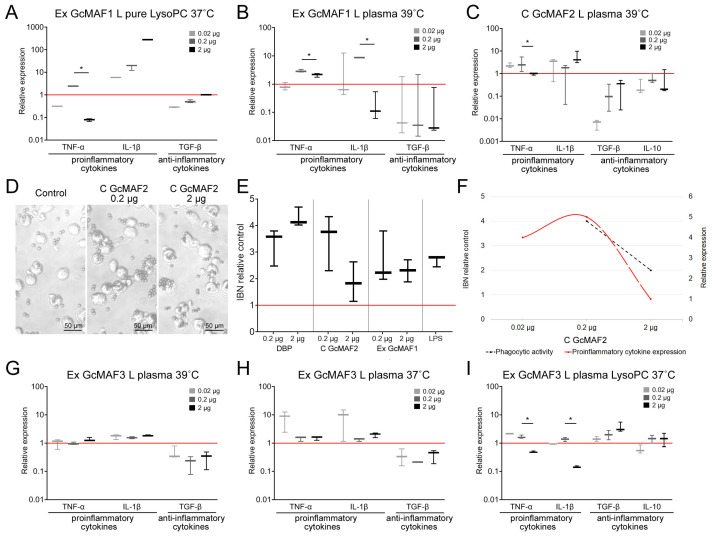
The effects of different GcMAF specimens derived from different donors using different methods on cytokine mRNA expression and phagocytic activity of murine PMs with 0.02 µg, 0.2 µg, and 2 µg preparation with respect to the control (untreated), whose expression level was assumed to be 1 (the red solid line). (**A**–**C**) mRNA expression of the cytokine genes in peritoneal macrophages of mice treated with GcMAF derived from different donors using different methods. (**A**) Ex GcMAF1 L pure LysoPC 37 °C. (**B**) Ex GcMAF1 L plasma 39 °C. (**C**) C GcMAF2 L plasma 39 °C. (**D**,**E**) Assessment of the phagocytic activity of peritoneal macrophages treated with C GcMAF2 according to their ability to internalize metallic beads. (**D**) Images of the beads phagocytized by naïve macrophages (control) and macrophages treated with C GcMAF2. (**E**) Quantification of phagocytic activity of PMs (IBN) treated with DBP, C GcMAF2, Ex GcMAF1, and LPS (10 µg/mL) as a positive control. The naïve macrophages whose phagocytic activity was assumed to be 1 were used as a control (red solid line). (**F**) Comparison of phagocytic activity (dashed line) and mRNA expression of pro-inflammatory cytokines (non-monotonic) in murine PMs activated with different doses of C GcMAF2. (**G**–**I**) mRNA expression of the cytokine genes in PMs of mice treated with GcMAF derived from the same donor (GcMAF 3) using different methods. (**G**) Ex GcMAF3 Lplasma 39 °C. (**H**) Ex GcMAF3 L plasma 37 °C. (**I**) Ex GcMAF3 L plasma LysoPC 37 °C. IL-10 is not shown in (**A**,**B**,**G**,**H**) due to undetectable level of corresponding mRNA in these samples. We noted significance of differences between GcMAF doses 0.2 µg and 2 µg (* *p* < 0.05; Mann–Whitney U test).

**Figure 5 ijms-24-17396-f005:**
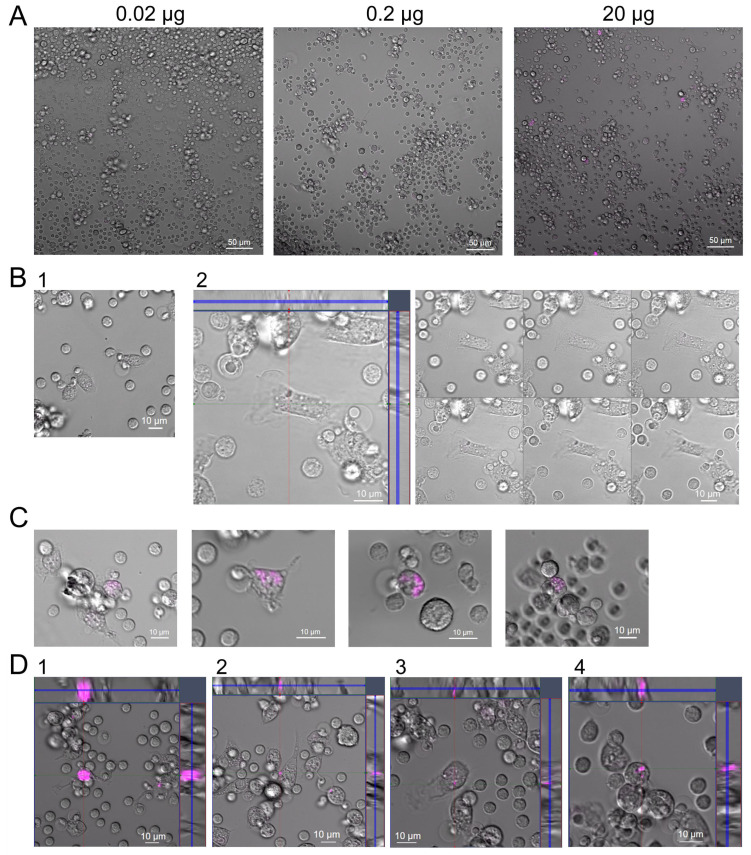
Fluorescent cytology analysis of the interaction between PMs and Cy5-GcMAF contained in the incubation medium at different doses. (**A**) The general view of PMs incubated in the presence of three Cy5-GcMAF doses: 0.02 μg, 0.2 μg, and 20 μg. (**B**) PMs incubated in the presence of 0.02 μg ligand. A very small amount of Cy5-GcMAF is distributed in the cytoplasm of PMs. (**C**) PMs incubated in the presence of 0.2 μg ligand. Labeled Cy5-GcMAF material is clearly detected in the cytoplasm of PMs. (**D**) PMs incubated in the presence of 2 μg ligand. Two variants of distribution of the labeled Cy5-GcMAF material are detected. Cy5-GcMAF occupies most of the cell interior (1); Cy5-GcMAF resides on the external cell surface as a well-defined spot (2–4).

**Figure 6 ijms-24-17396-f006:**
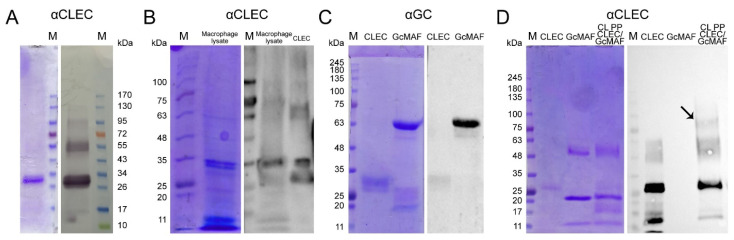
An analysis of the interaction between GcMAF and CLEC10A. (**A**) The interaction between CLEC10A protein and anti-CLEC10A antibodies in Western blotting. High-molecular-weight derivatives of CLEC10A protein are clearly visualized. Coomassie staining (left) and Western blot (right). (**B**) Western blotting of the interaction of PM lysate proteins and CLEC10A protein with anti-CLEC10A antibodies. Coomassie staining (left) and Western blot (right). (**C**) Western blotting of the cross-reactivity of anti-GcMAF antibodies and CLEC10A protein. Coomassie staining (left) and Western blot (right). (**D**) Analysis of the direct interaction between GcMAF and CLEC10A in the cross-link experiment using paraformaldehyde as a cross-linking agent and anti-CLEC10A antibodies. Coomassie staining (left) and Western blot (right). In the Western blot image, one can clearly see that there is no cross-homology between GcMAF and anti-CLEC10A antibodies. An arrow shows the cross-linked product, with molecular weight corresponding to that for the GcMAF/CLEC10A complex (lane 3).

**Figure 7 ijms-24-17396-f007:**
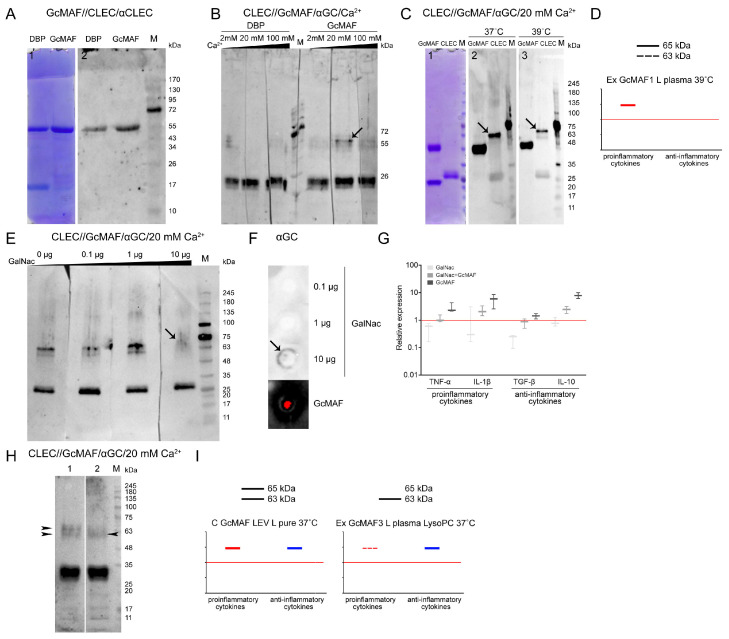
An analysis of direct interaction between GcMAF and CLEC10A. (**A**) Sandwich-type assay of direct interaction between fractions of the proteins isolated from 25-OH-D_3_/Sepharose^®^: DBP and ApoA1, as well as interaction between GcMAF in the same sample and CLEC10A protein. Coomassie staining (left) and Western blot (right). The GcMAF3 sample was used for analysis. The result indicates that both DBP and GcMAF3 bind directly to CLEC10A. ApoA1 does not directly interact with CLEC10A. (**B**) Sandwich-type assay demonstrating that Ca^2+^ ions are needed for direct interaction between high-molecular-weight CLEC10A and GcMAF derivatives. The results indicate that the high-molecular-weight CLEC10A derivative (~65 kDa) binds directly to GcMAF protein when the incubation Western blot system contains 20 mM Ca^2+^. CLEC10A protein does not bind to DBP precursor at selected Ca^2+^ concentrations. An arrow shows the specific signal. (**C**) An analysis of the effect of temperature on direct interaction between CLEC10A and GcMAF proteins. Sandwich-type assay. Coomassie staining (left) and Western blot (right). Arrows show the specific signal. The GcMAF3 sample is used for analysis. (**D**) An avatar of the diagram of mRNA expression of the analyzed cytokine genes in PMs treated with GcMAF1 (Figure 4A) demonstrating that the pro-inflammatory response of PMs depends on binding between GcMAF and the 65 kDa CLEC10A derivative. The control (untreated PMs) whose expression level was assumed to be 1 was denoted via the red solid line. (**E**) Sandwich-type dot blot assay of the interaction between CLEC10A and GcMAF under the conditions of competitive presence of free GalNAc. The results indicate that when the incubation Western blot system contains 10 µg of free GalNAc, the bond between the 65 kDa CLEC10A derivative and GcMAF breaks. An arrow shows degradation of the specific signal. The GcMAF3 sample is used for analysis. (**F**) An analysis of the cross-interaction between free GalNAc and anti-Gc antibodies. An arrow shows the specific signal of cross-interaction; (**G**) in vivo analysis of mRNA expression of the analyzed cytokine genes in PMs under the conditions of competitive interaction between GcMAF and free GalNAc. The GcMAF3 sample is used for analysis. The control (untreated PMs) whose expression level was assumed to be 1 was denoted via the red solid line. (**H**) Sandwich-type assay of the direct interaction between CLEC10A and two GcMAF variants (lane 1—C GcMAF LEV L pure 37 °C; lane 2—GcMAF3 L plasma LysoPC 37 °C) characterized by opposite abilities to activate PMs toward mRNA expression of pro- and anti-inflammatory cytokines. Arrows show changes in the interaction between the high-molecular-weight CLEC10A derivatives (63 kDa and 65 kDa) and GcMAF. In the variant where GcMAF exhibits stronger pronounced anti-inflammatory properties (GcMAF3 L plasma LysoPC 37 °C), it is mostly the high-molecular-weight (63 kDA) CLEC10A derivative that specifically interacts with GcMAF. (**I**) Avatars of the diagrams of mRNA expression of the analyzed cytokine genes in PMs treated with C GcMAF LEV L pure 37 °C and GcMAF3 L plasma LysoPC 37 °C (Figure 3B and Figure 4A, respectively). The control (untreated PMs) whose expression level was assumed to be 1 was denoted via the red solid line. The graphs (D,I) schematically show: a red line indicates the direction of the pro-inflammatory response, a blue line indicates the direction of the anti-inflammatory response. Black lines indicate CLEC10A derivatives. Solid lines indicate strong interaction of GcMAF with the CLEC10A derivatives and a strong inflammatory response. Dashed lines indicate weak interaction of GcMAF with the CLEC10A derivatives and a weak inflammatory response. The absence of a line in the graph means that GcMAF does not interact with the CLEC10A derivatives. The main idea of the graph is to show that the interaction of GcMAF with the 65 kDa CLEC10A derivative induces a pro-inflammatory response, and with the 63 kDa derivative an anti-inflammatory response.

**Figure 8 ijms-24-17396-f008:**
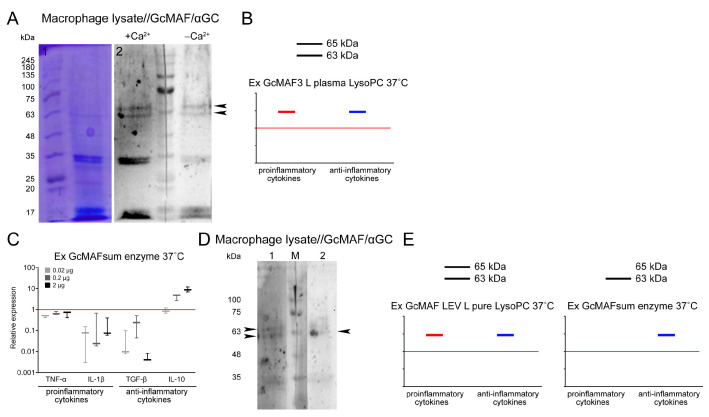
An analysis of direct interaction between GcMAF and PM lysate proteins. (**A**) Sandwich-type assay of the direct interaction between PM lysate and GcMAF protein in the presence and absence of Ca^2+^ ions. Coomassie staining (left) and sandwich-type Western blot assay (right). The blot is found to contain specifically reacting protein fractions corresponding to the conventional 29 kDa CLEC10A and two high-molecular-weight fractions whose mobilities correspond to those of the high-molecular-weight CLEC10A derivatives. The absence of Ca^2+^ ions in the incubation system of sandwich-type Western blot assay significantly reduces the intensity of the specific signal characterizing the direct interaction between the two proteins. In the absence of Ca^2+^ ions, the 63 kDa band reacts with GcMAF to a smaller extent than with the 65 kDa protein. Both fractions interact with GcMAF with high efficiency in the presence of Ca^2+^ ions. (**B**) Avatars of the diagrams of mRNA synthesis of the analyzed cytokines in PMs treated with Ex GcMAF3 L plasma LysoPC 37 °C (Figure 4I). The control (untreated PMs) whose expression level was assumed to be 1 was denoted via the red solid line (**C**) The diagram of mRNA synthesis of the analyzed cytokines in PMs treated with the GcMAF preparation produced by conversion using purified enzymes. The control (untreated PMs) whose expression level was assumed to be 1 was denoted via the red solid line. (**D**) Sandwich-type assay of the direct interaction of PM lysate with Ex GcMAF LEV L pure LysoPC 37 °C (differently oriented cytokine mRNA synthesis) (lane 1) and Ex GcMAFsum enzyme 37 °C (the pronounced anti-inflammatory vector of mRNA synthesis of the analyzed cytokines) (lane 2). In the former case, both high-molecular-weight CLEC10A derivatives interact with GcMAF. The 63 kDa derivative predominantly binds to GcMAF in the latter case. (**E**) Avatars of the diagrams of mRNA synthesis of the analyzed cytokines in PMs treated with Ex GcMAF LEV L pure LysoPC 37 °C and Ex GcMAFsum enzyme 37 °C. The control (untreated PMs) whose expression level was assumed to be 1 was denoted via the red solid line. The graphs (**B**,**E**) schematically show: a red line indicates the direction of the pro-inflammatory response, a blue line indicates the direction of the anti-inflammatory response. Black lines indicate CLEC10A derivatives. Solid lines indicate strong interaction of GcMAF with the CLEC10A derivatives and a strong inflammatory response. The absence of a line in the graph means that GcMAF does not interact with the CLEC10A derivatives. The main idea of the graph is to show that the interaction of GcMAF with the 65 kDa CLEC10A derivative induces a pro-inflammatory response, and with the 63 kDa derivative an anti-inflammatory response.

**Figure 9 ijms-24-17396-f009:**
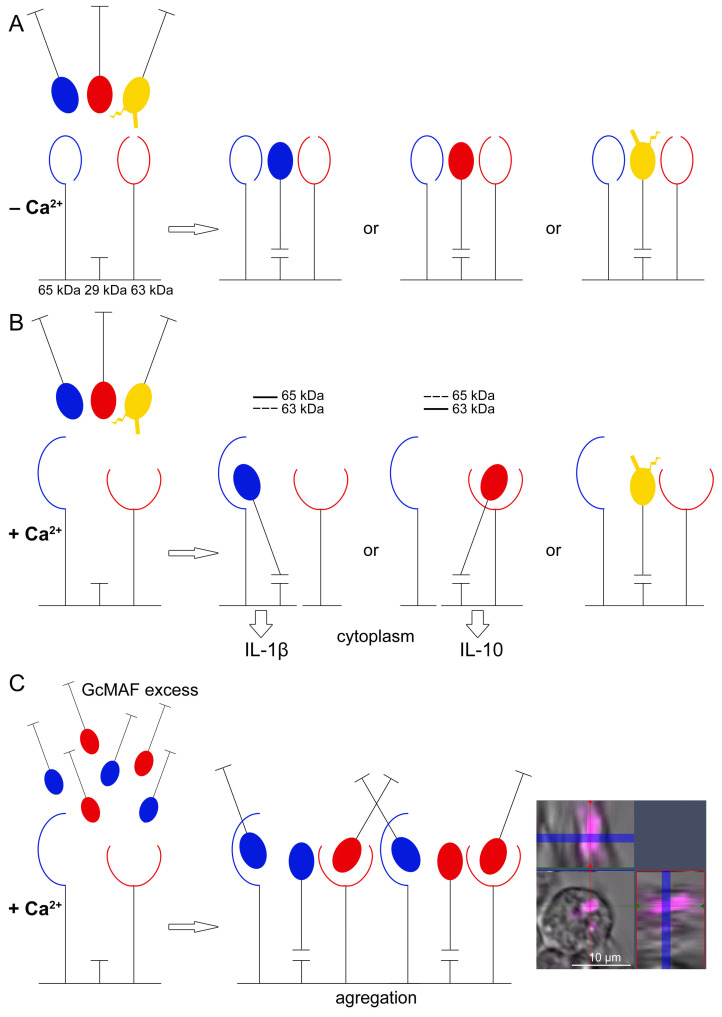
The schematic diagram of the presumable CLEC10A–GcMAF interaction and mRNA synthesis of the analyzed cytokines. (**A**,**B**) CLEC10A resides on the surface of PMs in C57BL/6 mice as three derivatives interacting with anti-CLEC10A antibodies: the canonical 29 kDa protein and two high-molecular-weight derivatives (~63 kDa and ~65 kDa). The three receptor derivatives seemingly form a single co-functioning complex. This conclusion has been drawn according to the findings demonstrating that in sequential experiments with the same preparation, C GcMAF LEV L pure 37 °C, mRNA of predominantly pro- or anti-inflammatory cytokines, or both of them simultaneously can be synthesized. The CLEC10A 29 kDa derivative is the primary “anchor” that fixes the DBP/GcMAF molecule on the cytoplasmic membrane, regardless of the presence or absence of Ca^2+^ molecules. CLEC10A 29 kDa can bind only one DBP/GcMAF molecule. In this regard, there is competition for binding to the primary “anchor”. In the absence of Ca^2+^, this contact apparently leads to nonspecific stochastic synthesis of the mRNA of the analyzed cytokines. With a sufficient concentration of Ca^2+^, epitopes of specific Ca^2+^-dependent high-molecular derivatives of the CLEC10A 63/65 kDa receptor “open”. Depending on which GcMAF molecule binds to the primary “anchor”, its secondary binding by a modified sugar residue occurs with a derivative of either 63 kDa or 65 kDa. In both cases, the activated complex is internalized by the cytoplasm. mRNA synthesis of anti-inflammatory and pro-inflammatory cytokines is induced in the former and latter cases, respectively. (**C**) Competitive interactions between the “anchoring” 29 kDa derivative and the 63/65 kDa high-molecular-weight derivatives occur in the presence of ligand excess and Ca^2+^ ions. It is assumed that complexes are aggregated when the complex between the “anchoring” 29 kDa derivative and 63 kDa/65 kDa derivatives cannot be formed and engagement of high-molecular-weight Ca^2+^-dependent derivatives of the CLEC10A receptor and GcMAF takes place. The absence of conditions required for functional complex formation and its internalization in the cytoplasm inhibits the synthesis of both pro- and anti-inflammatory cytokines. DBP is indicated in yellow; blue and red are two variants of the GcMAF-molecule-binding derivatives 65 kDa and 63 kDa CLEC10A, respectively.

**Table 1 ijms-24-17396-t001:** The sequences of primers used in this study (for—a forward primer; rev—a reverse primer).

Primers	Oligonucleotide Sequences
TNF-α-for	5′-AAGCCTGTAGCCCACGTCGTA-3′
TNF-α-rev	5′-GGCACCACTAGTTGGTTGTCTTTG-3′
IL-1β-for	5′-TCCAGGATGAGGACATGAGCAC-3′
IL-1β-rev	5′-GAACGTCACACACCAGCAGGTTA-3′
TGF-β-for	5′-GTGTGGAGCAACATGTGGAACTCTA-3′
TGF-β-rev	5′-TTGGTTCAGCCACTGCCGTA-3′
IL-10-for	5′-GACCAGCTGGACAACATACTGCTAA-3′
IL-10-rev	5′-GATAAGGCTTGGCAACCCAAGTAA-3′
GAPDH-for	5′-AAATGGTGAAGGTCGGTGTG-3′
GAPDH-rev	5′-TGAAGGGGTCGTTGATGG-3′

## Data Availability

The data supporting the findings of this study are available from the corresponding author upon reasonable request.

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
