# Peer review of "The Molecular Aspects of Functional Activity of Macrophage-Activating Factor GcMAF"

_ijms, 2023, doi:10.3390/ijms242417396_

Round 1

Reviewer 1 Report

Comments and Suggestions for Authors

Comments to the Authors:

While the manuscript by Kirikovich et al. is commendable for investigating the functional activity of GcMAF on peritoneal macrophages, several deficiencies hinder the comprehension of results. To improve readability, the following revisions are recommended:

Provide statistical significance in the figures to facilitate data interpretation, especially considering the absence of a specific trend in a dose-dependent manner.

Address discrepancies in Figure 2, such as the mismatch between the number of doses in Fig.2B legend and the actual data, as well as the absence of IL-10 data in Fig.2C and 2E. Ensure visibility of the color legend for the lowest dose in all figures using distinct grey shades.

Clarify the experimental design, including the inclusion of an untreated group and confirmation of relative expression. Explain the differently oriented inflammatory response mentioned on Line 235, particularly in relation to the expression of TGF-β in the lowest dose.

Include statistical significance asterisks between groups to facilitate data comparison. Reconsider the axis scale in the figures, as most mean values are below 10, making a scale up to 100 unnecessary and hindering the observation of differences between groups.

In Figure 3F, explain the lack of effect of LPS on phagocytosis and clarify the non-lethality of 10μg LPS to cells. Provide the duration of the treatment.

Address missing data in Figure 3D, where the highest dose is absent, and IL-10 expression is not provided in Figures 3A and 3D.

Clarify the discrepancy in Line 326, where it is mentioned that GCMAF1 elicits a proinflammatory response, but Figure 4A shows a decrease in TNFα and an increase in IL-1β. Additionally, clarify the significance of these changes in Figure 4B.

Confirm the correct citation of figures, as mentioned in Line 328, ensuring consistency between the cited figure and the actual reference (e.g., Fig.4C instead of 3C).

Author Response

Dear Reviewer,

We appreciate you for your precious time in reviewing our paper and providing valuable comments. It was your valuable and insightful comments that led to possible improvements in the current version. We have carefully considered the comments and tried our best to address every one of them. We hope the manuscript after careful revisions meet your high standards. We welcome further constructive comments if any. Below we provide the point-by-point responses. All modifications in the manuscript have been highlighted in yellow.

Sincerely, Dr. Sergey Bogachev and PhD Svetlana Kirikovich

Reviewer 2 Report

Comments and Suggestions for Authors

1.     Introduction: Although the authors are attempting to evoke a certain feeling here, with the greater and growing "distrust" between scientists and the general public, I would recommend the authors select an alternative phrasing than "arousing mistrust among research community".

2.     Figure 2: Figure legend and figure don't match but are assumed to be equivalent between 1x/0.02, 10x/0.2, and 100x/2 μg. Please include the dose in a normalized format for all data in the manuscript.

3.     Figure 3: metallic beads are both located in and supposedly inside of the macrophages. Please confirm that the beads are internalized and not locationed on top of or stuck onto the macrophage. Also, please specify the mode of internalization for metallic beads versus organic substrates and if there is any concern or thought to this mode of action. If actually internalized, macrophages appear to be saturated with beads so there is unlikely to be an accurate dose response mechanism present between beads and macrophages.

4.     Figure 3G: The authors attempt to show correlation between phagocytic activity and anti-infmallatory cytokine expression. Phagocytosis is an innate immune response which in the presence of foreign bodies (like metallic beads) will likely cause the release of pro-inflammatory cytokines. It would be critical to understand the effect of these in parallel with the anti-inflammatory cytokines to see if there is also a similar trend of those or an opposite. This reviewer recommends the authors analyze some select mRNA expression of preferred pro-inflammatory cytokines and to plot accordingly as well. Also, please describe the apparent trend decreaseing from 0.2 to 1-2 μg.

5.     Dome-shape of the mRNA expression is suggested to be accurately referred to as a non-monototic or non-hormetic dose response curve throughout.

6. Overall, the manuscript is very difficult to follow and includes a lot of data. There is not a discussion or conclusion section here that can help provide a clear thought.  The authors would benefit greatly from re-formatting the paper into a traditional format with conclusions. I also recommend a summary figure because there is too much data to see what it all comes together like. 

Author Response

(The authors gave the same response as above.)

Reviewer 3 Report

Comments and Suggestions for Authors

The presented data in the manuscript “The Molecular Aspects of Functional Activity of Macrophage-Activating Factor GcMAF” of Kirikovich et al. deals about different methods to obtain different proteins of DBP of one and other different donors out of blood.

New is the aspect that different chromatic isolations result in DBP with or without regions of actin and/or vitamin D binding sites. Furthermore, the team used different forms in activation of DBP proteins to GcMAF.

The usage of these different activated forms results in pro- and anti-inflammatory activation of different used cytokines and interleukins’.

New is also the aspect that GcMAF possess the ability to bind to ApoAI and that GcMAF binds to the protein C part of a lectin-receptor dependent of Calcium concentrations.

It is very hard to read all presented data because there is an overload of information because of the very detailed isolation of Vitamin D Binding proteins (low and high molecular masses), its activation by different forms etc.

May it is better to make two manuscripts, one for isolation of DBP and its activation forms and secondly binding properties to lectin receptors (incl. Apo lipoprotein A etc.).

Additionally, it was found that GcMAF does activate macrophages, some reports show that the binding of Cholecalciferol to GcMAF increased the activation of macrophages more than the GcMAF alone and stimulate further cytotoxic cells, like dendritic cells and natural killer cells of healthy donors.

One more information is needed if the media contains vitamin D derivatives or not (FCS media) which may affect the results of the pro-and anti-inflammatory aspects on PMs and cells.

Line 177, the first sentence is not a full sentence, please go through the whole manuscript about these errors.

Comments on the Quality of English Language

Some minor errors are in the manuscript,

e.g. line 177, the sentence is not clear.

Author Response

(The authors gave the same response as above.)

Round 2

Reviewer 1 Report

Comments and Suggestions for Authors

Thank you for addressing the comments.

Reviewer 2 Report

Comments and Suggestions for Authors

thank you for addressing my comments. I have no further questions.